# A General Framework for Off-Policy Learning with Partially-Observed Reward

**Rikiya Takehi**
Waseda University
rikiya.takehi@fuji.waseda.jp

**Masahiro Asami**
HAKUHODO Technologies Inc.
masahiro.asami@hakuhodo-technologies.co.jp

**Kosuke Kawakami**
HAKUHODO Technologies Inc.
kosuke.kawakami@hakuhodo-technologies.co.jp

**Yuta Saito**
Cornell University
ys522@cornell.edu

## Abstract

*Off-policy learning* (OPL) in contextual bandits aims to learn a decision-making policy that maximizes the target rewards by using only historical interaction data collected under previously developed policies. Unfortunately, when rewards are only partially observed, the effectiveness of OPL degrades severely. Well-known examples of such partial rewards include explicit ratings in content recommendations, conversion signals on e-commerce platforms that are partial due to delay, and the issue of censoring in medical problems. One possible solution to deal with such partial rewards is to use secondary rewards, such as dwelling time, clicks, and medical indicators, which are more densely observed. However, relying solely on such secondary rewards can also lead to poor policy learning since they may not align with the target reward. Thus, this work studies a new and general problem of OPL where the goal is to learn a policy that maximizes the expected target reward by leveraging densely observed secondary rewards as supplemental data. We then propose a new method called **Hy**brid Policy Optimization for **P**artially-Obs**e**rved **R**eward (**HyPeR**), which effectively uses the secondary rewards in addition to the partially-observed target reward to achieve effective OPL despite the challenging scenario. We also discuss a case where we aim to optimize not only the expected target reward but also the expected secondary rewards to some extent; counter-intuitively, we will show that leveraging the two objectives is in fact advantageous also for the optimization of only the target reward. Along with statistical analysis of our proposed methods, empirical evaluations on both synthetic and real-world data show that HyPeR outperforms existing methods in various scenarios.

## 1 Introduction

In applications that involve dynamic decision-making, such as ad placement and recommendation systems, exploration can be costly and risky. These constraints limit the use of online exploration of actions, thereby motivating the study of offline policy learning methods. **Off-policy learning (OPL)** addresses this need by enabling the policy optimization using only logged bandit data generated under old (or *logging*) policies (Joachims et al., 2018; Su et al., 2019; 2020a; Uehara et al., 2022).

A typical approach to OPL is using policy gradient (PG), which we can estimate unbiasedly based on the logged data via techniques like Inverse Propensity Score (IPS) and Doubly Robust (DR) (Dudík et al., 2014). OPL methods based on PG iterations perform effectively in ideal scenarios with large sample sizes and relatively small action spaces (Saito & Joachims, 2021; Sachdeva et al., 2023; Taufiq et al., 2024). However, in many application scenarios, rewards are only partially observed due to missing data (Jakobsen et al., 2017; Wang et al., 2019b; Jadidinejad et al., 2019; Christakopoulou et al., 2022), censoring (Ren et al., 2019; Wang et al., 2019a), delayed observation (Wang et al., 2022b; Imbens et al., 2022; Wang et al., 2023; Saito et al., 2024a), data fusion (Imbens et al., 2022), and multi-stage rewards (Wan & McAuley, 2018; Hadash et al., 2018; Ma et al., 2018; Saito, 2020). For instance, on e-commerce platforms, binary conversions serve as a target reward to maximize.

Table 1: Examples of Partially-Observed Rewards in Various Real-Life Scenarios

| Scenario | Reward | Reason for Partial Observation |
|---|---|---|
| Streaming Platform | Explicit Rating | Multi-stage & Missing (Hadash et al., 2018) |
| E-commerce | Conversion | Multi-stage & Delayed (Ktena et al., 2021) |
| Ad-placement | Product Purchase | Missing & Delayed (Ashkan & Clarke, 2009) |
| Clinical Trials | Long-term Health | Censoring (Pratap et al., 2020) |
| Financial Investment | Long-term Return | Delayed (Arroyo et al., 2019) |
| Subscription Platform | User Retention | Delayed (Wang et al., 2022b) |
| Remediation | Academic Performance | Delayed (Barnett, 2011) |

Unfortunately, conversions are only observed for products that were clicked and seen by the user. To make matters worse, conversion signals are obtained only after weeks-long delays (Ktena et al., 2021), and most of them are yet to be observed when policy learning is conducted. Table 1 provides a list of real-life examples and causes of partially-observed rewards. When rewards are only partially observed due to such causes, the estimation of PG would result in high variance and inefficient OPL.

Our work thus proposes a new formulation and method to address this challenging but highly general problem: **OPL with partial observations of the reward**. In scenarios with only partial observations of the target reward, one possible idea to perform OPL more effectively is to use more frequently observed *secondary rewards* instead. For instance, in e-commerce recommender systems, we often observe not only the conversion signals but also other implicit feedback such as clicks and dwell time, which are much more densely observed with no missingness (Liu et al., 2010; Jadidinejad et al., 2019). Thus, in our problem formulation, we consider both the *target reward* that is only partially observed (e.g., user ratings, conversions, retention, future earnings, and survival days) and the *secondary rewards* that are fully observed (e.g., clicks, dwell time, and short-term medical indicators). Considering the case where we also aim to optimize secondary rewards, we explore learning a policy that maximizes a weighted sum of the target reward and secondary rewards. This represents a more general objective rather than focusing only on the optimization of the target reward.

With the availability of the secondary rewards, one feasible approach to perform OPL is to maximize some aggregation of the secondary rewards as a surrogate for the target reward. Nonetheless, a significant drawback of this approach is the potential for high PG estimation bias, as secondary rewards often do not align perfectly with the target reward (Liu et al., 2010; Jadidinejad et al., 2019). We also do not know how to construct an appropriate aggregation function of the secondary rewards. Therefore, there is a dilemma between the target and secondary rewards: the former more accurately aligns with our ultimate objective, while the latter provides more observations. Using only the former leads to high variance in PG estimation due to missing observations, while relying solely on the latter results in high bias due to a potential misalignment with the ultimate objective.

To solve this new OPL problem more effectively, we focus on developing a method that can leverage both target and secondary rewards in a principled way. Specifically, we propose **Hy**brid Policy Optimization for **P**artially-Obs**e**rved **R**eward (HyPeR), a method which estimates the PG leveraging both types of rewards. We show that our PG estimator can substantially reduce the estimation variance compared to typical estimators such as IPS and DR while maintaining unbiasedness under reasonable conditions. In addition, while our approach generally performs effectively using the predefined weight between the target and secondary rewards within the objective, we demonstrate that further enhancement can be achieved by strategically *tuning* the weight to improve the bias-variance trade-off of the PG estimation, which we can perform based only on observable logged data. Finally, we conduct comprehensive experiments on both synthetic and real-world datasets, where the HyPeR algorithm outperforms a range of existing methods in terms of optimizing both the target reward objective and the combined objective of the target and secondary rewards.

The key contributions of our work can be summarized as follows.

- We formulate the general problem of Off-Policy Learning (OPL) for contextual bandits with partially-observed rewards, encompassing many prevalent scenarios such as missing data, delayed rewards, and censoring, all of which are instances of this general problem.
- We propose a new method to address OPL with partial rewards by leveraging more densely observed secondary rewards to estimate the policy gradient with reduced variance.
- We consider a combined objective defined by a weighted sum of the target and secondary rewards and introduce the novel concept of a strategic use of an incorrect weight in our method to maximize the advantage of combining the two types of rewards.

## 2 OFF-POLICY LEARNING

### 2.1 THE CONVENTIONAL FORMULATION

We first formulate the conventional OPL problem regarding the typical contextual bandit setting (Dudík et al., 2014; Swaminathan & Joachims, 2015a; Farajtabar et al., 2018). In the typical formulation, context $x \in \mathcal{X} \subseteq \mathbb{R}^{d_x}$ is a $d_x$-dimensional vector that is drawn i.i.d. from an unknown distribution $p(x)$. Given context $x$, a possibly stochastic policy $\pi(a|x)$ chooses an action $a$ within a finite action space denoted here by $\mathcal{A}$. The (target) reward $r \in [0, r_{max}]$ (e.g., ratings, conversions, retention, survivals) is then sampled from an unknown conditional distribution $p(r|x, a)$.

The existing literature defines the objective function in OPL by the expected target reward (often referred to as the *policy value*) as below (Swaminathan & Joachims, 2015b).

$$V(\pi) := \mathbb{E}_{p(x)\pi(a|x)p(r|x,a)}[r] = \mathbb{E}_{p(x)\pi(a|x)}[q(x,a)], \tag{1}$$

where $q(x, a) := \mathbb{E}[r \mid x, a]$ is the expected reward given $x$ and $a$, which we call the *q-function*. In OPL, the goal is to learn a policy $\pi_\theta$, parameterized by $\theta$, that would maximize the policy value: $\theta^* \in \arg\max_{\theta \in \Theta} V(\pi_\theta)$. In particular, OPL aims to learn such a policy using only logged data consisting of tuples $(x, a, r)$ generated under the *logging* policy denoted by $\pi_0$. More specifically, the logged data we can use to perform OPL can be written as $\mathcal{D} = \{(x_i, a_i, r_i)\}_{i=1}^n \sim p(\mathcal{D})$ where the data distribution is induced by the logging policy, i.e., $p(\mathcal{D}) = \prod_{i=1}^n p(x_i)\pi_0(a_i|x_i)p(r_i|x_i, a_i)$.

### 2.2 OFF-POLICY LEARNING VIA POLICY GRADIENT

Most existing approaches in OPL are based on PG iterations (Ma et al., 2020; Chen et al., 2021). This method updates the policy parameter $\theta$ via iterative gradient ascent: $\theta_{t+1} \leftarrow \theta_t + \nabla_\theta V(\pi_\theta)$ where the policy gradient is represented as $\nabla_\theta V(\pi_\theta) = \mathbb{E}_{p(x)\pi_\theta(a|x)}[q(x,a)\nabla_\theta \log \pi_\theta(a|x)]$ Saito et al. (2024b). This form of PG can be derived by the log-derivative trick and suggests to update the policy parameter $\theta$ so that the resulting policy $\pi_\theta$ can choose actions that have high expected reward.

To implement the PG iteration, we first need to estimate the PG, since it is an unknown vector. To achieve this using only the logged data $\mathcal{D}$ collected under the logging policy $\pi_0$, the relevant literature relies on estimators such as IPS and DR, which enable unbiased estimation of the PG (Dudík et al., 2014; Metelli et al., 2021). These PG estimators are specifically defined as follows.

$$\nabla_\theta \widehat{V}_{\text{IPS}}(\pi_\theta; \mathcal{D}) := \frac{1}{n} \sum_{i=1}^n w(x_i, a_i) r_i g_\theta(x_i, a_i), \tag{2}$$

$$\nabla_\theta \widehat{V}_{\text{DR}}(\pi_\theta; \mathcal{D}) := \frac{1}{n} \sum_{i=1}^n \left\{ w(x_i, a_i)(r_i - \hat{q}(x_i, a_i))g_\theta(x_i, a_i) + \mathbb{E}_{\pi_\theta(a|x_i)}[\hat{q}(x_i, a)g_\theta(x_i, a)] \right\}, \tag{3}$$

where $w(x, a) := \pi_\theta(a \mid x)/\pi_0(a \mid x)$ is the importance weight and $g_\theta(x, a) := \nabla_\theta \log \pi_\theta(a \mid x)$ is the policy score function. $\hat{q}(x, a)$ is an estimator of the q-function, which we can obtain by performing reward regression in the logged data $\mathcal{D}$.

These estimators are unbiased (i.e., $\mathbb{E}_{p(\mathcal{D})}[\nabla_\theta \widehat{V}_{\text{IPS}}(\pi_\theta; \mathcal{D})] = \mathbb{E}_{p(\mathcal{D})}[\nabla_\theta \widehat{V}_{\text{DR}}(\pi_\theta; \mathcal{D})] = \nabla_\theta V(\pi_\theta)$) under *full support*, which requires that the logging policy sufficiently explore the action space.

> **Condition 1** (Full Support). *The logging policy $\pi_0$ is said to have full support if $\pi_0(a|x) > 0$ for all $x \in \mathcal{X}$ and $a \in \mathcal{A}$.*

The importance weighting technique used by IPS and DR enables an unbiased estimation of the PG, thereby resulting in effective OPL even without additional exploration. However, if the problem is far from ideal with smaller data sizes, large action spaces, and noisy or partial rewards, existing OPL methods can easily collapse due to substantial variance in PG estimation (Saito & Joachims, 2022).

### 2.3 PARTIALLY-OBSERVED REWARDS

As discussed in the introduction and in Table 1, there are many real-life cases where we can observe the target reward *only partially*. To precisely formulate such a scenario, we introduce an additional random variable, $o \in \{0, 1\}$, to represent whether the reward is observed for each data point. If $o_i = 1$, the reward $r_i$ is observed; otherwise, it is unavailable, as indicated by setting $r_i = \text{N/A}$. Considering a scenario where the observation indicator comes from an unknown distribution $0 < p(o|x) < 1$, we can generalize the data generating process as $\mathcal{D} = \{(x_i, a_i, o_i, r_i)\}_{i=1}^n \sim p(\mathcal{D}) = \prod_{i=1}^n p(x_i)\pi_0(a_i|x_i)p(o_i|x_i)p(r_i|x_i, a_i)$, which encompasses many realistic situations like missing data, delayed rewards, data fusion, multi-stage rewards, and censoring.

To implement OPL with such partially-observed rewards, we can simply apply the PG approach using only the data with observed rewards, which we call **r-IPS** and **r-DR**:

$$\nabla_\theta \widehat{V}_{\text{r-IPS}}(\pi_\theta; \mathcal{D}) := \frac{1}{n} \sum_{i=1}^n \frac{o_i}{p(o_i|x_i)} w(x_i, a_i) r_i g_\theta(x_i, a_i), \tag{4}$$

$$\nabla_\theta \widehat{V}_{\text{r-DR}}(\pi_\theta; \mathcal{D}) := \frac{1}{n} \sum_{i=1}^n \frac{o_i}{p(o_i|x_i)} \left\{ w(x_i, a_i)(r_i - \hat{q}(x_i, a_i))g_\theta(x_i, a_i) + \mathbb{E}_{\pi_\theta(a|x_i)}[\hat{q}(x_i, a)g_\theta(x_i, a)] \right\}, \tag{5}$$

where we can see that these estimators use only the data with reward observations ($o_i = 1$) to estimate the PG. We can readily show that these estimators are unbiased, but they are often substantially inefficient and produce high variance in PG estimation. This is because they use only a part of the data in $\mathcal{D}$ and naively discard all the information when $o_i = 0$.

## 3 OFF-POLICY LEARNING WITH SECONDARY REWARDS

To deal with the issue of inefficiency in OPL when the rewards are partial, we propose a new formulation of OPL with secondary rewards. In many real-life scenarios, we not only observe the target reward such as the conversion signals that we are optimizing, but also secondary rewards such as clicks and dwell time (Wan & McAuley, 2018; Jadidinejad et al., 2019; Christakopoulou et al., 2022). By leveraging these secondary rewards, we aim to reduce variance in PG estimation and to achieve a more efficient OPL even under the challenging scenarios with partially-observed rewards.

To implement this idea, we extend the typical formulation of OPL by introducing the secondary rewards denoted by $s \in \mathbb{R}^{d_s}$, which can be multi-dimensional. We consider the secondary rewards to be sampled from an unknown conditional distribution of the form: $p(s|x, a)$ after taking action $a$ for context $x$. The logged dataset that we can use in the setting can be written as follows.

$$\mathcal{D} := \{(x_i, a_i, o_i, s_i, r_i)\}_{i=1}^n \sim p(\mathcal{D}) = \prod_{i=1}^n p(x_i)\pi_0(a_i|x_i)p(o_i|x_i)p(s_i|x_i, a_i)p(r_i|x_i, a_i, s_i), \tag{6}$$

In addition to the introduction of secondary rewards, we consider the generalized objective called the *combined policy value*, which is defined as a weighted sum of the expected target and secondary rewards, considering a situation where we aim to optimize the secondary rewards as well.

$$V_c(\pi; \beta) := (1 - \beta)V_r(\pi) + \beta V_s(\pi) = (1 - \beta)\mathbb{E}_{p(x)\pi(a|x)}[q(x, a)] + \beta\mathbb{E}_{p(x)\pi(a|x)} \left[ \sum_{d=1}^{d_s} f_d(x, a) \right], \tag{7}$$

where $f_d(x, a) := \mathbb{E}[s_d|x, a]$ is the $d$-th dimension of the expected secondary reward and $\beta \in [0, 1)$ is a parameter to control the prioritization between the optimization of the target reward and that of the secondary rewards. When $\beta = 0$, this generalized objective reduces to the typical policy value defined in Eq. (1). With some positive value of $\beta$, we can address practical situations where we aim to optimize not only the target reward but also the secondary rewards to some extent.

Given this extended problem of OPL with secondary rewards and with the combined policy value, we can consider a baseline method of using some aggregation of the secondary rewards as a surrogate

for the target reward. This baseline approach of using only the secondary rewards as surrogates, which we call **s-IPS** and **s-DR**, estimates the PG as follows:

$$\nabla_\theta \hat{V}_{\text{s-IPS}}(\pi_\theta; \mathcal{D}) = \frac{1}{n} \sum_{i=1}^{n} w(x_i, a_i) F(s_i) g_\theta(x_i, a_i), \tag{8}$$

$$\nabla_\theta \hat{V}_{\text{s-DR}}(\pi_\theta; \mathcal{D}) \tag{9}$$

$$= \frac{1}{n} \sum_{i=1}^{n} \left\{ w(x_i, a_i)(F(s_i) - F(\hat{f}(x_i, a_i))) g_\theta(x_i, a_i) + \mathbb{E}_{\pi_\theta(a|x_i)}[F(\hat{f}(x_i, a)) g_\theta(x_i, a)] \right\},$$

where $\hat{f}(x, a)$ is an estimator of $f(x, a)$ and $F(s)$ is some aggregation of the secondary rewards, such as their weighted average, designed to imitate the target reward. By replacing the target reward with $F(s)$, s-IPS and s-DR can use all the data points irrespective of the observation indicator $o_i$, thus reducing the variance. However, unless the function $F(s)$ accurately describes the target reward, which is untestable, the estimators often produce substantial bias against the true PG regarding the target reward, which leads to ineffective OPL as we will show in our experiments.

## 4 HYBRID POLICY OPTIMIZATION FOR PARTIALLY-OBSERVED REWARD

This section proposes a new OPL algorithm to maximize the combined policy value in Eq. (7) with only partially-observed target rewards and secondary rewards that do not necessarily align accurately with the target reward. We also newly introduce the concept of strategically using a different value of the balancing factor $\gamma$ compared to the true value of $\beta$ in the objective in Eq. (7) to improve the finite-sample effectiveness of the algorithm.

The key idea behind our algorithm is the introduction of a new estimator of the PG using secondary rewards to reduce variance while also optimizing the secondary rewards depending on the value of $\beta$ in the objective function.

To derive our method, we first focus on the PG estimation regarding the target policy value $V_r(\pi_\theta)$. Particularly to address the variance issue in existing methods, we propose leveraging secondary rewards through the following estimator for $\nabla_\theta V_r(\pi_\theta)$.

$$\nabla_\theta \hat{V}_r(\pi_\theta; \mathcal{D}) := \frac{1}{n} \sum_{i=1}^{n} \left\{ \mathbb{E}_{\pi_\theta(a|x_i)} [\hat{q}(x_i, a) g_\theta(x_i, a)] + w(x_i, a_i) \left( \hat{q}(x_i, a_i, s_i) - \hat{q}(x_i, a_i) \right) g_\theta(x_i, a_i) \right.$$

$$\left. + \frac{o_i}{p(o_i|x_i)} w(x_i, a_i) \left( r_i - \hat{q}(x_i, a_i, s_i) \right) g_\theta(x_i, a_i) \right\}, \tag{10}$$

where $\hat{q}(x, a, s)$ is an estimator of the q-function conditional on the secondary rewards (i.e., $q(x, a, s)$), which we can derive, e.g., by solving $\hat{q} = \arg\min_{q'} \sum_{(x, a, s, r) \in \mathcal{D}} (r - q'(x, a, s))^2$ using $\mathcal{D}$. Note also that we can estimate the conditional reward-observation probability $p(o_i|x_i)$ when it is unknown by regressing $o$ to $x$ observed in the logged data $\mathcal{D}$ by a supervised classifier. We need to perform this estimation task before performing OPL not only for our method, but also for baseline methods such as r-IPS and r-DR.

We first show that our PG estimator is unbiased under the same conditions as r-DR.

> **Theorem 1.** *(Unbiasedness) Under Condition 1, Eq. (10) is unbiased against the true PG regarding the target reward, i.e.,*
>
> $$\mathbb{E}[\nabla_\theta \hat{V}_r(\pi_\theta; \mathcal{D})] = \nabla_\theta V_r(\pi_\theta) = \nabla_\theta V_c(\pi_\theta; \beta = 0)$$
>
> *See Appendix B.1 for the proof.*

In addition to its unbiasedness, our PG estimator fully utilizes the information from the secondary rewards, thus reducing the variance compared to r-DR in most cases. The following demonstrates that the variance of Eq. (10) can be much lower than r-DR.

**Theorem 2.** *(Variance Reduction) Under Condition 1, we have*

$$n(\mathbb{V}_{\mathcal{D}}[\nabla_\theta \hat{V}_{\text{r-DR}}(\pi; \mathcal{D})] - \mathbb{V}_{\mathcal{D}}[\nabla_\theta \hat{V}_r(\pi; \mathcal{D})])$$

$$= \mathbb{E}_{p(x)\pi_0(a|x)p(s|x,a)}\left[\frac{\rho^2}{p(o|x)^2}w(x,a)^2 g_\theta(x,a)^2 \left(\Delta_{q,\hat{q}\neg s}(x,a,s)^2 - \Delta_{q,\hat{q}}(x,a,s)^2\right)\right]$$

*where $\rho^2 := \mathbb{V}[o|x]$ is the variance of the observation indicator. $\Delta_{q,\hat{q}\neg s}(x,a,s) :=$ $q(x,a,s) - \hat{q}(x,a)$ and $\Delta_{q,\hat{q}}(x,a,s) := q(x,a,s) - \hat{q}(x,a,s)$ are the estimation error of $\hat{q}(x,a)$ and $\hat{q}(x,a,s)$, respectively. See Appendix B.2 for the proof.*

Theorem 2 indicates that there is a reduction in variance if $\hat{q}(x,a,s)$ is better than $\hat{q}(x,a)$ in estimating $q(x,a,s)$, which is often the case since secondary rewards are typically somewhat correlated with the target reward. Thus, Eq. (10) is expected to perform better than r-DR while being unbiased under the same condition. To demonstrate this, in the experimental sections, we investigate the difference in performance between Eq. (10), which we name **HyPeR($\gamma = 0$)**, against r-DR.

Based on the new estimator for the PG regarding the target reward defined in Eq. (10), we finally introduce our **HyPeR** estimator.

$$\nabla_\theta \hat{V}_{\text{HyPeR}}(\pi_\theta; \mathcal{D}, \gamma) = (1 - \gamma) \cdot \nabla_\theta \hat{V}_r(\pi_\theta; \mathcal{D}) + \gamma \cdot \nabla_\theta \hat{V}_s(\pi_\theta; \mathcal{D}), \tag{11}$$

where the first term $\nabla_\theta \hat{V}_r(\pi_\theta; \mathcal{D})$ estimates the PG regarding the target policy value $V_r(\pi_\theta)$ as in Eq. (10). The second term $\nabla_\theta \hat{V}_s(\pi_\theta; \mathcal{D})$ estimates the PG regarding the secondary policy value $V_s(\pi_\theta)$, which we can define by simply applying DR to the sum of secondary rewards as

$$\nabla_\theta \hat{V}_s(\pi_\theta; \mathcal{D})$$

$$= \frac{1}{n}\sum_{i=1}^n \left\{\mathbb{E}_{\pi_\theta(a|x_i)}\left[\sum_{d=1}^{d_s} \hat{f}_d(x_i,a)g_\theta(x_i,a_i)\right] + w(x_i,a_i)\sum_{d=1}^{d_s}\left(s_i^d - \hat{f}_d(x_i,a_i)\right)g_\theta(x_i,a_i)\right\}.$$

In Eq. (11), $\gamma \in [0,1)$ is a tunable parameter that defines the mixture ratio between the estimators of the PG regarding the target reward and secondary rewards. When using the predefined weight (i.e., $\gamma = \beta$), Eq. (11) is unbiased against the combined policy gradient (i.e., $\mathbb{E}[\nabla_\theta \hat{V}_{\text{HyPeR}}(\pi_\theta; \mathcal{D})] = \nabla_\theta V(\pi_\theta)$), but the following discusses a strategic tuning of $\gamma$ to further improve the finite-sample effectiveness of our HyPeR algorithm.

## 4.1 STRATEGIC TUNING OF THE WEIGHT $\gamma$

Although the natural choice of $\gamma$ in our PG estimator is $\beta$, which defines the true objective in Eq. (7), we argue that *intentionally using $\gamma \neq \beta$ can further improve the combined policy value in the finite-sample setting*. This is due to the potential differences in variance between the different types of rewards. Specifically, strategically shifting weights from the predefined balance, $\beta$, can lead to better estimation due to variance reduction, even though it introduces some bias. For example, consider an objective function that is a sum of two estimands, $X + Y$. If estimator $\hat{X}$ carries less variance than estimator $\hat{Y}$, it is likely better to give more weight to $\hat{X}$ to achieve less variance, even at the cost of introducing some bias. When dealing with multiple reward types, this can occur in various situations, such as when one reward is less noisy than the other, or as in our study, when one reward type (secondary reward) is more frequently observed than the other (target reward).

In HyPeR, the PG regarding the target reward $\nabla_\theta \hat{V}_r(\pi_\theta; \mathcal{D})$ often carries much higher variance compared to that regarding the secondary rewards $\nabla_\theta \hat{V}_s(\pi_\theta; \mathcal{D})$ due to the partial-observation nature. Thus, although setting $\gamma = \beta$ will make the HyPeR estimation unbiased and at least reduce the variance from existing methods, it can still have high variance, particularly when $\beta$ takes a small value. On the other hand, increasing the weight of $\gamma$ will likely lead to less variance since this prioritizes the second term more, while introducing some bias. This creates an interesting bias-variance trade-off, and we aim to tune $\gamma$ to achieve the optimal combined policy value by the resulting policy:

$$\gamma^* = \arg\max_{\gamma \in [0,1)} V(\pi_\theta(\cdot; \gamma, \mathcal{D}); \beta), \tag{12}$$

where $\pi_\theta(\cdot; \gamma, \mathcal{D})$ is a policy optimized under the weight $\gamma$ used in our policy gradient estimator in Eq. (11), and its value is evaluated under the originally defined weight $\beta$.

Since the true policy value is unknown, we need to perform Eq. (12) from only the logged dataset $\mathcal{D}$. The most straightforward method is via splitting the dataset $\mathcal{D}$ into training ($\mathcal{D}_{tr}$) and validation ($\mathcal{D}_{val}$) sets. Then, we train a policy using $\mathcal{D}_{tr}$ (i.e., $\pi_\theta(\cdot; \gamma, \mathcal{D}_{tr})$) and estimate the value using $\mathcal{D}_{val}$ (i.e., $\hat{V}(\pi_\theta; \beta, \mathcal{D}_{val})$). However, an issue with this naive procedure is that the policy $\pi_\theta(\cdot; \gamma, \mathcal{D}_{tr})$ is trained on a smaller dataset $\mathcal{D}_{tr}$ instead of the full dataset $\mathcal{D}$ that is used in real training. In the PG estimation, variance is inversely proportional to the data size $|\mathcal{D}|$, while bias is unaffected. Therefore, the naive tuning procedure may end up selecting a higher weight than the optimal weight $\gamma^*$ by overly prioritizing variance reduction. To address this issue, we ensure that $|\mathcal{D}_{tr}| = |\mathcal{D}|$ through bootstrapping:

$$\mathcal{D}'_{tr} = \left\{ (x'_i, a'_i, s'_i, o'_i, r'_i) \mid i = 1, ..., n \right\}, \tag{13}$$

where $\{(x'_i, a'_i, s'_i, o'_i, r'_i)\}_{i=1}^n \sim \mathcal{D}_{tr}$ is sampled independently *with replacement*. By doing so, we alleviate the issue regarding the variance estimation, since $\mathcal{D}'_{tr}$ now has the same size as the full dataset $\mathcal{D}$. Using $\mathcal{D}'_{tr}$, we propose to solve the bi-level optimization to data-drivenly tune $\gamma$:

$$\hat{\gamma}^* = \arg\max_{\gamma \in [0,1]} \hat{V}(\pi_\theta(\cdot; \gamma, \mathcal{D}'_{tr}); \beta, \mathcal{D}_{val}). \tag{14}$$

## 5 SYNTHETIC EXPERIMENT

**Synthetic Data Generation.** To create synthetic data, we sample 10-dimensional context vectors $x$ from a standard normal distribution, and the sample size is fixed at $n = 2000$ by default. We then synthesize the logging policy as $\pi_0 = \text{softmax}(\phi(x^T \mathcal{M}_{X,A} a + x^T \theta_x + a^T \theta_a))$, where $\mathcal{M}, \theta_x, \theta_a$ are parameter matrices randomly sampled from a uniform distribution with range $[-1, 1]$, and actions $a \in \mathcal{A}$ are sampled following this logging policy ($|\mathcal{A}| = 10$). $\phi$ is a parameter that controls how deterministic the logging policy would become, and we set this at $\phi = -2.0$ by default. We then synthesize each dimension of the 5-dimensional expected secondary rewards given $x$ and $a$ as

$$f_d(x, a) = x^T \mathcal{M}'_{X,A} a + x^T \theta'_x + a^T \theta'_a, \tag{15}$$

and secondary rewards $s$ are sampled from a normal distribution $s_d \sim \mathcal{N}(f_d(x, a), \sigma_s^2)$. In the main text, we set the default to $\sigma_s = 0.5$, and the results for other values of $\sigma_s$ can be found in Appendix C.2. We finally synthesize the expected target reward function as

$$q(x, a, f(x, a)) := (1 - \lambda)(x^T \mathcal{M}''_{X,A} a + x^T \theta''_x + a^T \theta''_a + x^T \mathcal{M}_{X,F} f + a^T \mathcal{M}_{A,F} f) + \lambda f^T \theta_f. \tag{16}$$

$\lambda \in [0, 1]$ is an experimental parameter to control how much secondary rewards are correlated with the target reward. When $\lambda = 1$, secondary rewards are completely correlated with the target reward, while a smaller $\lambda$ will make it less correlated. We use $\lambda = 0.7$ as a default setting throughout the synthetic experiment. The target reward is sampled from a normal distribution as $r \sim \mathcal{N}(q(x, a, f(x, a)), \sigma_r^2)$ with default $\sigma_r = 0.5$ (results with other $\sigma_r$ values are provided in Appendix C.2). The target reward observation probability is an experimental parameter and is set to $p(o|x) = 0.2$ for all $x$ by default.[1] The true weight $\beta$, which is used to define the combined policy value $V_c(\pi; \beta)$, is set to $\beta = 0.3$ and it is also one of the experimental parameters.

In the synthetic experiments, we generally use the predefined weight $\beta$ for HyPeR, and represent it as HyPeR($\gamma = \beta$). We compare HyPeR($\gamma = \beta$) against HyPeR($\gamma = 0$), r-IPS, r-DR, s-IPS and s-DR. For s-IPS and s-DR, we use $F(s) = s^T(\theta_f + \varepsilon_F)$, where $\theta_f$ is identical to that of Eq. (16), and $\varepsilon_F \sim \mathcal{N}(0, \sigma_F^2)$ is the noise to simulate the inaccuracy of describing the target reward.

### 5.1 RESULTS AND DISCUSSION

We run OPL simulations 100 times with different train-test splits. We report the relative policy values calculated by $(V(\pi^*) - V(\pi_\theta))/(V(\pi_\theta) - V(\pi_{\text{unif}}))$, where $\pi^*$ is the optimal policy and $\pi_{\text{unif}}$ is a uniform random policy. This way, the performance of a random policy will have a value of 0, and an optimal policy would have a value of 1, thus easier to interpret. Note that the shaded regions in the plots represent 95% confidence intervals estimated via bootstrapping.

---

[1]Appendix C.2 presents additional experimental results with estimated observation probabilities, which support the same conclusion as the main text.

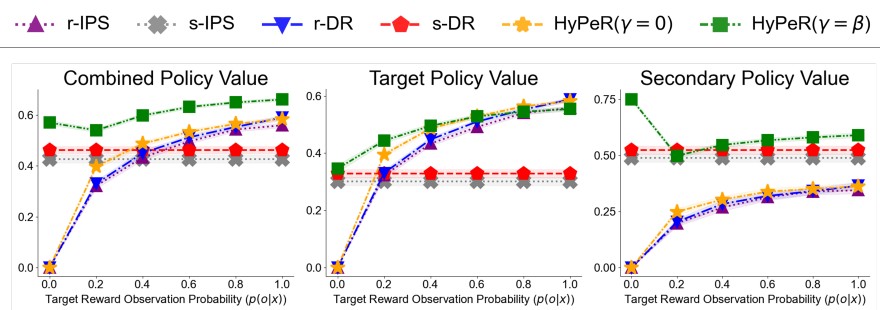

Figure 1: Comparing the combined, target, and secondary policy values of OPL methods with **varying target reward observation probabilities** ($p(o|x)$) on Synthetic Data.

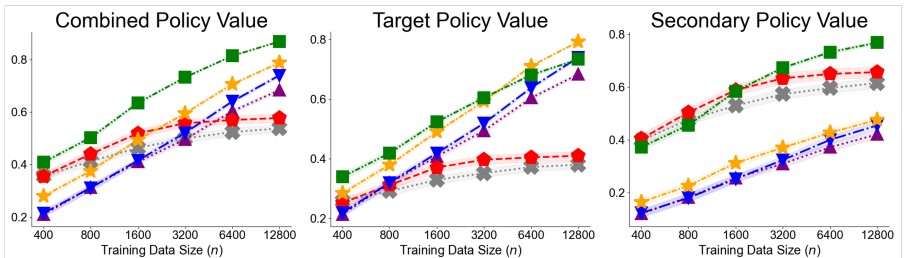

Figure 2: Comparing the combined, target, and secondary policy values of OPL methods with **varying training data sizes** ($n$) on Synthetic Data.

**How does HyPeR perform with varying target reward observation probabilities?**   Figure 1 evaluates the relative policy value when we vary the observation probability of the target reward $p(o|x)$. We observe that HyPeR($\gamma = \beta$) provides the highest combined policy value in all cases. It is also able to optimize both the target and secondary policy values in a balanced way. In addition, HyPeR($\gamma = \beta$) outperforms all the baselines regarding the target policy value, and it even outperforms HyPeR($\gamma = 0$) when the observation probability is low. This is because the secondary reward maximization component of HyPeR reduces the variance of the PG estimation, as secondary rewards are denser. We also observe that HyPeR($\gamma = 0$) (target reward maximization component of HyPeR) is always performing better than r-DR, which suggests the effective use of secondary rewards enhances the estimation of the target gradient $\nabla_\theta V_r(\pi_\theta)$ alone, as in Theorem 2.

**How does HyPeR perform with varying training data sizes?**   Figure 2 evaluates the methods' policy values with different training data sizes. Larger data size generally makes gradient estimations more accurate as it decreases the variance. Therefore, as the data size increases, the methods that use the target reward in their estimation (i.e., HyPeR($\gamma = \beta$), HyPeR($\gamma = 0$), r-IPS, and r-DR) perform increasingly better compared to the ones that do not (i.e., s-IPS and s-DR). The left plot in Figure 2 shows that HyPeR($\gamma = \beta$) performs the best in most cases regarding the combined policy value $V_c(\pi)$. It even performs the best in terms of the target policy value $V_r(\pi)$, particularly when the data size is small due to the use of secondary rewards and resulting variance reduction.

**How does HyPeR perform with varying correlation between target and secondary rewards?** Figure 3 shows the results with different degrees of correlation between the secondary rewards and the target reward, controlled by $\lambda$ in Eq. (16). A larger $\lambda$ indicates that the target reward can be more predictable by the secondary reward. We can also see that HyPeR($\gamma = \beta$) is the best option in almost all cases for both the combined policy value and target policy value. We also observe that HyPeR($\gamma = \beta$), s-IPS, and s-DR perform comparatively well when secondary rewards are more correlated (larger $\lambda$), as they find more advantage in using the secondary rewards as surrogates.

**How do HyPeR and data-driven weight selection perform with varying true weight $\beta$?**   Section 4.1 showed that a data-driven tuning of the weight $\gamma$ will potentially improve the performance of HyPeR. To empirically evaluate whether our weight-tuning method can make further improvement, in addition to HyPeR($\gamma = \beta$), we add HyPeR(Tuned $\hat{\gamma}^*$) and HyPeR(Optimal $\gamma^*$) for comparison.

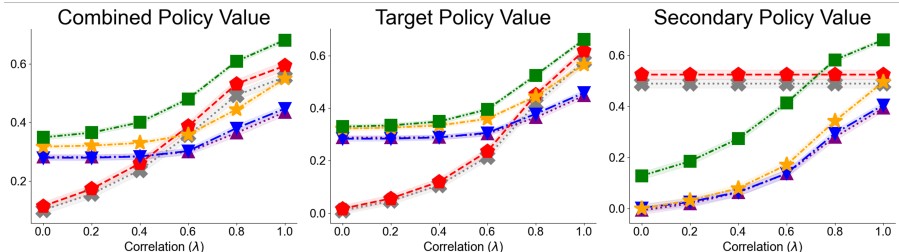

Figure 3: Comparing the combined, target, and secondary policy values of OPL methods with **varying degrees of target-secondary reward correlation** ($\lambda$) on Synthetic Data. Secondary rewards can completely explain the target reward at $\lambda = 1$, and they are less correlated with smaller $\lambda$.

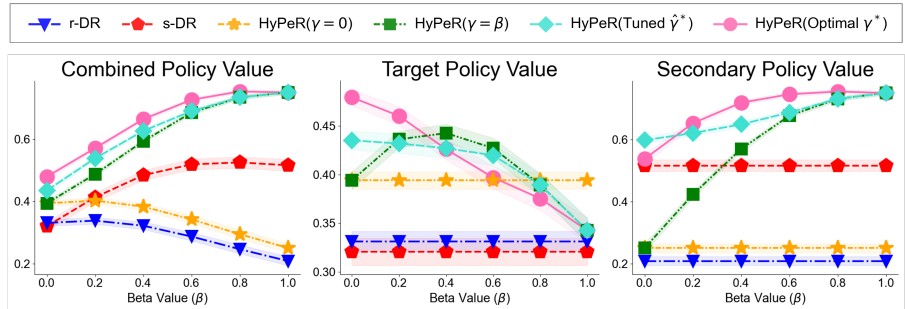

Figure 4: Comparing the combined, target, and secondary policy values of OPL methods with **varying $\beta$**, which is a weight that balances target and secondary policy values in the combined policy value. Higher $\beta$ means the secondary policy value becomes more dominant.

HyPeR(Tuned $\hat{\gamma}^*$) estimates the optimal weight through estimation using the method described in Section 4.1. HyPeR(Optimal $\gamma^*$) is a skyline that performs HyPeR with a truly optimal weight $\gamma^*$, which is not feasible in practice but it is a useful reference. Due to the increase in the number of comparisons, here we reduce r-IPS and s-IPS from the baselines, which always perform worse than DR-based methods (Appendix C.1 shows the complete results including r-IPS and s-IPS). Figure 4 provides the results, including HyPeR(Tuned $\hat{\gamma}^*$) and HyPeR(Optimal $\gamma^*$), with varying weight $\beta$. From the results, we can see that HyPeR(Tuned $\hat{\gamma}^*$) always outperforms HyPeR($\gamma = \beta$) and all other feasible methods, suggesting the effectiveness of our weight tuning procedure. This observation also interestingly implies that an incorrect weight (i.e., $\hat{\gamma}^* \neq \beta$) can lead to a better policy performance compared to $\gamma = \beta$, due to the variance reduction at the cost of some bias in the PG estimation as discussed in Section 4.1. Appendix C.1 also empirically demonstrates the advantage of leveraging the bootstrapping procedure in our tuning process.

## 6 REAL-WORLD EXPERIMENT

To assess the real-world applicability of HyPeR, we now evaluate it on the KuaiRec dataset (Gao et al., 2022).[2] This is a publicly available fully-observed user-item matrix data collected on a short video platform, where 1,411 users have viewed all 3,317 videos and left watch duration as feedback. This unique feature of KuaiRec enables to perform an OPL experiment without synthesizing the reward function (few other public datasets retain this desirable feature).

**Setup.** We use watch ratio (= watch duration/video length) as target reward $r$. We use four-dimensional secondary rewards; each dimension has a realistic reason why the platform would want to maximize it. The first dimension is binary, where $s_1 = 1$ if $r \geq 2.0$, and $s_1 = 0$ if $r < 2.0$; this reward maximization lets the platform prioritize sessions with an exceptionally long watch ratio. The second dimension is also binary, where $s_2 = -1$ if $r < 0.5$, and $s_2 = 0$ if $r \geq 0.5$. This dimension is built to strictly punish and refrain from sessions with an exceptionally low watch ratio (raising the engagement floor). The third dimension of the secondary rewards is time since video upload (multiplied by -1), which is implemented to prioritize newer videos over older ones. The last

---

[2]https://kuairec.com/

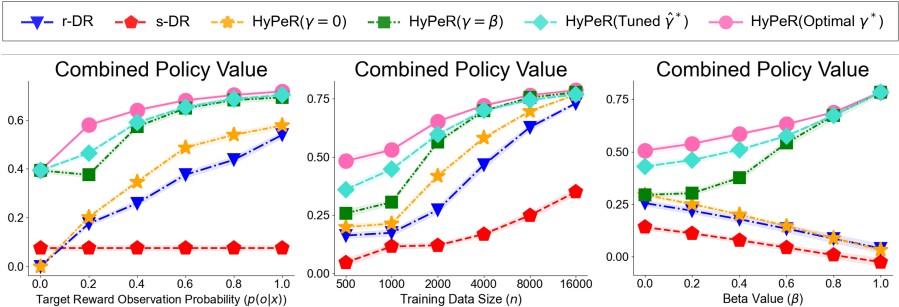

Figure 5: Comparing the **combined policy values** of OPL methods under (**left**) varying target reward observation probabilities ($p(o|x)$), (**center**) varying data sizes ($n$), and (**right**) varying true weights $\beta$ in the combined policy value on the **KuaiRec** dataset.

dimension is video length, which is designed to prioritize longer videos when watch ratios are similar. Note that all continuous rewards are normalized to the range $[0, 1]$. We use the first dimension of the secondary rewards to express $F(s)$ for s-IPS and s-DR.

To perform OPL experiments on the dataset, we randomly choose 988 users (70%) for training and 423 users (30%) for evaluation. We set the target reward observation probability to $p(o|x) = 0.2$ for all $x$, training data size to $n = 1000$, and weight $\beta = 0.3$, as default experimental parameters. The actions are chosen randomly with size $|\mathcal{A}| = 100$, and Appendix C.3 shows results with varying numbers of actions. We define the logging policy as $\pi_0(a|x) = \mathrm{softmax}(\phi(x^T \mathcal{M}_{X,A} a + x^T \theta_x + a^T \theta_a))$, with $\phi = -2.0$, and run 100 simulations with different train-test splits

**Results.** Figure 5 provides real-world experiment results with varying target reward observation probabilities $p(o|x)$, varying training data sizes $n$, and varying weights $\beta$ in the combined policy value. In this section, we provide comparisons of the combined policy values for r-DR, s-DR, HyPeR($\gamma = 0$), HyPeR($\gamma = \beta$), HyPeR(Tuned $\hat{\gamma}^*$) and HyPeR(Optimal $\gamma^*$). In Figure 5, we observe that HyPeR($\gamma = \beta$) and HyPeR(Tuned $\hat{\gamma}^*$) both outperform the baselines by far. Moreover, by comparing HyPeR(Tuned $\hat{\gamma}^*$) against HyPeR($\gamma = \beta$), we can see that intentionally using the tuned weight leads to better performance, particularly under more challenging scenarios (i.e., sparse target reward observation and small training data size). We also observe, in Appendix C.3, that HyPeR outperforms the baseline methods in terms of not only the combined policy value, but also the target and secondary policy values. It is also interesting to see that, in terms of the secondary policy value, the policy performance becomes much worse when we use only the target reward (i.e., performance of r-DR at $\beta = 1$), compared to the case where we use only the secondary rewards (i.e., performance of HyPeR($\gamma = \beta$) at $\beta = 1$). This demonstrates that the secondary rewards that we used in our real experiment is not highly correlated with the target reward, ensuring the problem is non-trivial and making it crucial to leverage both types of rewards via our hybrid approach.

# 7  CONCLUSION AND FUTURE WORK

This paper studies the general problem of off-policy learning (OPL) in contextual bandits with partially-observed target rewards. Relying only on the target reward to learn a policy often leads to a high variance and inefficient learning in this setup. We propose **Hybrid Policy Optimization for Partially-Observed Reward** (**HyPeR**), a novel approach that integrates secondary rewards to improve policy gradient estimation. HyPeR effectively reduces variance while maintaining unbiasedness and can easily be extended to deal with the combined objective of maximizing both the target and secondary rewards from available data. Experiments on synthetic and real-world datasets demonstrate that HyPeR outperforms existing methods in optimizing both the target and combined policy values. In future work, it would be valuable to extend our formulation and method to the problem of offline reinforcement learning. Conducting an experiment in a real environment would also further demonstrate the effectiveness, applicability, and real-world impact of our method.

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

# A    RELATED WORK

## A.1    OFF-POLICY LEARNING

The contextual bandit framework has been used as one of the favored methods for online decision-making under uncertainty (Lattimore & Szepesvári, 2020). There is also a growing demand for refining decision-making using only historical datasets without requiring active exploration. Consequently, off-policy learning (OPL) methods, which learn decision-making policies solely from historical datasets collected under a different policy in the contextual bandit framework, have attracted attention (Joachims et al., 2018). This area of research could be applied to many real-world interactive systems like recommender systems and ad placement (Saito & Joachims, 2021; Aminian et al., 2024).

There are mainly two families of approaches in OPL: regression-based approach and policy-based approach. The regression-based method uses a regression estimate that was trained on the data to predict the rewards from the logged data (Sachdeva et al., 2020; Jeunen & Goethals, 2021). The policy is then made by simply choosing the action with the highest predicted reward deterministically or from a distribution based on the estimates. A drawback of this approach is the bias formed by the inaccurate regression estimation. In contrast, the policy-based approach aims to update the policy by gradient ascent iterations. The policy gradient needs to be estimated from data using techniques like IPS and DR (Dudík et al., 2014; Su et al., 2020a). These estimators are unbiased under certain assumptions. However, when rewards are only partially observed, they often suffer from extremely high variance (Saito & Joachims, 2021; Sachdeva et al., 2023; Taufiq et al., 2024). One possible approach to address this variance issue is through the use of secondary rewards that are more frequently observed instead, which lowers variance substantially. However, as secondary rewards often misalign with the target rewards, this suffers greatly from bias. Targeting this problem, HyPeR effectively uses both types of rewards to enhance the estimation of the gradient.

We did not compare pessimistic OPL methods in our experiments because they are not relevant to our context (Swaminathan & Joachims, 2015a; London & Sandler, 2019; Gabbianelli et al., 2024; Sakhi et al., 2024). Our main motivation is to address the prevalent problem of partial target rewards, an issue that pessimistic techniques are not designed to solve. However, our proposed method could easily be combined with a pessimistic approach if desired. It is also important to note that when tuning hyperparameters in OPL, there is the issue of overestimation in the validation estimate of the policy value. In such cases, it may be beneficial to use correction methods as proposed in (Saito & Nomura, 2024). Additionally, the line of work on adaptive estimator selection (Su et al., 2020b; Udagawa et al., 2023) is relevant in this context. However, their primary focus is on data-adaptive estimator selection for OPE, rather than on effectively tuning hyperparameters for OPL.

Our work is also related to the multi-objective optimization problems (Wang et al., 2022a; Alizadeh et al., 2023). However, these works solely focus on achieving weights that meet the designer's desired balance of multiple rewards, where the rewards generally have no relation to each other. In contrast, our work is crucially different from the fact that we use secondary rewards to *enhance* the estimation of the target rewards and the combined reward function, particularly in a practical situation where target rewards are only partially observed.

## A.2    PARTIAL TARGET REWARDS AND SECONDARY REWARDS

Our work is related to various fields that deal with partial observations of rewards due to reasons like missing data (Jakobsen et al., 2017; Wang et al., 2019b; Jadidinejad et al., 2019; Christakopoulou et al., 2022), censoring (Ren et al., 2019; Wang et al., 2019a), delayed observation (Wang et al., 2022b; 2023; Saito et al., 2024a), data fusion (Imbens et al., 2022), and multi-stage rewards (Wan & McAuley, 2018; Hadash et al., 2018). Problems involving partially-observed data have been extensively studied. For example, Liu et al. (2010) propose to unify implicit and explicit feedback in a recommendation setting, to enhance collaborative filtering. Wan & McAuley (2018) have addressed the implicit-explicit feedback problem by modeling them into a monotonic behavior chain and considering them as multi-stage rewards. Christakopoulou et al. (2022) have proposed to design a reinforcement learning model that uses user trajectories (secondary reward) to maximize user satisfaction (target reward). Wang et al. (2022b) investigate the relationship between short-term surrogates and long-term user satisfaction.

In the field of off-policy evaluation and learning, perhaps the closest setting to our work is LOPE (Saito et al., 2024a), which aims to estimate the long-term performance of the evaluation policy. To achieve this, in addition to the long-term rewards from historical datasets, Saito et al. (2024a) leverage short-term rewards collected in an *online* experiment. However, while Saito et al. (2024a) specifically focus on such a setting where short-term online experiments can be done, we do not assume access to such online experiment data. In addition, under their formulation, the two types of rewards have no difference in observation density, i.e., the long-term reward is not partial.

## B PROOFS AND DERIVATIONS

### B.1 PROOF OF THEOREM 1

*Proof.* To demonstrate the unbiasedness of the estimator in Eq. (10), we show that its expectation is equal to the true policy gradient given in Eq. (1).

$$\mathbb{E}_{\mathcal{D}}[\nabla_\theta \hat{V}_r(\pi_\theta; D)] = \mathbb{E}_{\mathcal{D}}\left[\frac{1}{n}\sum_{i=1}^n \mathbb{E}_{\pi_\theta(a|x_i)}\left[\hat{q}(x_i, a)g_\theta(x_i, a)\right]\right.$$

$$\left. + \frac{\pi_\theta(a_i|x_i)}{\pi_0(a_i|x_i)}\left(\hat{q}(x_i, a_i, s_i) - \hat{q}(x_i, a_i)\right)g_\theta(x_i, a_i) + \frac{o_i}{p(o|x)}\frac{\pi_\theta(a_i|x_i)}{\pi_0(a_i|x_i)}\left(r_i - \hat{q}(x_i, a_i, s_i)\right)g_\theta(x_i, a_i)\right]$$

$$= \mathbb{E}_{p(x)}\left[\mathbb{E}_{\pi_\theta(a|x)}\left[\hat{q}(x, a)g_\theta(x, a)\right]\right.$$

$$+ \mathbb{E}_{\pi_0(a|x)p(s|x,a)}\left[\frac{\pi_\theta(a|x)}{\pi_0(a|x)}\left(\hat{q}(x, a, s) - \hat{q}(x, a)\right)g_\theta(x, a)\right]$$

$$\left. + \mathbb{E}_{\pi_0(a|x)p(s|x,a)p(o|x)p(r|x,a,s)}\left[\frac{o}{p(o|x)}\frac{\pi_\theta(a|x)}{\pi_0(a|x)}\left(r - \hat{q}(x, a, s)\right)g_\theta(x, a)\right]\right]$$

$$= \mathbb{E}_{p(x)}\left[\mathbb{E}_{\pi_\theta(a|x)}\left[\hat{q}(x, a)g_\theta(x, a)\right] + \mathbb{E}_{\pi_0(a|x)}\left[\frac{\pi_\theta(a|x)}{\pi_0(a|x)}\left(\hat{q}(x, a, f(x, a)) - \hat{q}(x, a)\right)g_\theta(x, a)\right]\right.$$

$$\left. + \mathbb{E}_{\pi_0(a|x)p(o|x)}\left[\frac{o}{p(o|x)}\frac{\pi_\theta(a|x)}{\pi_0(a|x)}\left(q(x, a, f(x, a)) - \hat{q}(x, a, f(x, a))\right)g_\theta(x, a)\right]\right]$$

$$= \mathbb{E}_{p(x)}\left[\mathbb{E}_{\pi_\theta(a|x)}\left[\hat{q}(x, a)g_\theta(x, a)\right] + \mathbb{E}_{\pi_\theta(a|x)}\left[\left(\hat{q}(x, a, f(x, a)) - \hat{q}(x, a)\right)g_\theta(x, a)\right]\right.$$

$$\left. + \mathbb{E}_{\pi_0(a|x)}\left[\frac{\pi_\theta(a|x)}{\pi_0(a|x)}\left(q(x, a, f(x, a)) - \hat{q}(x, a, f(x, a))\right)g_\theta(x, a)\right]\right]$$

$$= \mathbb{E}_{p(x)\pi_\theta(a|x)}\left[q(x, a, f(x, a))\right] = \nabla_\theta V_r(\pi)$$

$\square$

## B.2 PROOF OF THEOREM 2

*Proof.* Using the law of total variance, the difference in variance of the $j$-th element of the gradient estimator can be decomposed into

$$
n(\mathbb{V}_{\mathcal{D}}[\nabla_\theta \hat{V}_{\text{r-DR}}(\pi_\theta; D)] - \mathbb{V}_{\mathcal{D}}[\nabla_\theta \hat{V}_{\text{r}}(\pi_\theta; D)]])
$$

$$
= \mathbb{E}_{p(x)\pi_0(a|x)p(s|x,a)p(o|x)}\left[ \left( \frac{o}{p(o|x)}w(x,a)g_\theta^{(j)}(x,a)\right)^2 \sigma^2(x,a,s)\right] \tag{17}
$$

$$
+ \mathbb{V}_{p(x)\pi_0(a|x)p(s|x,a)p(o|x)}\left[\mathbb{E}_{p(r|x,a,s)}\left[\mathbb{E}_{\pi_\theta(a|x)}[\hat{q}(x,a)g_\theta^{(j)}(x,a)]\right.\right.
$$

$$
\left.\left. + \frac{o}{p(o|x)}w(x,a)(r - \hat{q}(x,a))g_\theta^{(j)}(x,a)\right]\right]
$$

$$
- \mathbb{E}_{p(x)\pi_0(a|x)p(s|x,a)p(o|x)}\left[ \left( \frac{o}{p(o|x)}w(x,a)g_\theta^{(j)}(x,a)\right)^2 \sigma^2(x,a,s)\right]
$$

$$
- \mathbb{V}_{p(x)\pi_0(a|x)p(s|x,a)p(o|x)}\left[\mathbb{E}_{p(r|x,a,s)}\left[\mathbb{E}_{\pi_\theta(a|x)}[\hat{q}(x,a)g_\theta^{(j)}(x,a)]\right.\right.
$$

$$
\left.\left. + w(x,a)(\hat{q}(x,a,s) - \hat{q}(x,a))g_\theta^{(j)}(x,a) + \frac{o}{p(o|x)}w(x,a)(r - \hat{q}(x,a,s))g_\theta^{(j)}(x,a)\right]\right],
$$

where $\sigma^2(x,a,s) := \mathbb{V}[r|x,a,s]$, and $g_i^{(j)}(x,a)$ is the policy score function of the $j$-th element. We now focus on each of the terms. After canceling out the first and the third term, the second term further simplifies using the law of total variance:

$$
\mathbb{E}_{p(x)\pi_0(a|x)p(s|x,a)}\left[\frac{\rho^2}{p(o|x)^2}w^2(x,a)g_\theta^{(j)}(x,a)^2 \Delta_{q,\hat{q}\neg s}(x,a,s)^2\right] \tag{18}
$$

$$
+ \mathbb{V}_{p(x)\pi_0(a|x)p(s|x,a)}\left[\mathbb{E}_{\pi_\theta(a|x)}[\hat{q}(x,a)g_\theta^{(j)}(x,a)] + w(x,a)(q(x,a,s) - \hat{q}(x,a))g_\theta^{(j)}(x,a)\right],
$$

where $\rho^2 := \mathbb{V}[o|x]$, and $\Delta_{q,\hat{q}\neg s}(x,a,s) := q(x,a,s) - \hat{q}(x,a)$. The fourth term can be simplified into

$$
- \left(\mathbb{E}_{p(x)\pi_0(a|x)p(s|x,a)}\left[\frac{\rho^2}{p(o|x)^2}w^2(x,a)g_\theta^{(j)}(x,a)^2 \Delta_{q,\hat{q}}(x,a,s)^2\right]\right.
$$

$$
\left. + \mathbb{V}_{p(x)\pi_0(a|x)p(s|x,a)}\left[\mathbb{E}_{\pi_\theta(a|x)}[\hat{q}(x,a)g_\theta^{(j)}(x,a)] + w(x,a)(q(x,a,s) - \hat{q}(x,a))g_\theta^{(j)}(x,a)\right]\right)
$$

where $\Delta_{q,\hat{q}}(x,a,s) := q(x,a,s) - \hat{q}(x,a,s)$. Thus, we finally have

$$
n(\mathbb{V}_{\mathcal{D}}[\nabla_\theta \hat{V}_{\text{r-DR}}(\pi_\theta; D)] - \nabla_\theta \hat{V}_{\text{r}}(\pi_\theta; D)]) \tag{19}
$$

$$
= \mathbb{E}_{p(x)\pi_0(a|x)p(s|x,a)}\left[\frac{\rho^2}{p(o|x)^2}w(x,a)^2 g_\theta(x,a)^2 \left(\Delta_{q,\hat{q}\neg s}(x,a,s)^2 - \Delta_{q,\hat{q}}(x,a,s)^2\right)\right]
$$

$$\square$$

## C ADDITIONAL EXPERIMENTS

### C.1 DETAILED SYNTHETIC EXPERIMENT RESULTS

In this section, we provide more detailed experimental results for the comparisons conducted in Section 5. Specifically, we present results for additional baseline methods, including what we refer to as **r-DM** and **s-DM**. These are regression-based methods (also known as Direct Methods), which use regression estimates to predict expected target rewards based solely on logging data. The policy is then derived by choosing the action with the highest predicted reward in a deterministic manner. Note that r-DM trains its regression model on a partial dataset due to the use of target rewards, whereas s-DM trains on the full dataset but uses only secondary rewards for regression.

For the experiment involving weight-tuned methods (experiments with varying $\beta$ values), we also compare the values of parameters $\hat{\gamma}^*$ and $\gamma^*$ used for HyPeR (Tuned $\hat{\gamma}^*$) and HyPeR (Optimal $\gamma^*$).

Table 2: Comparison of the tuned $\gamma$ values for varying $\beta$ values

| $\beta$ | Mean $\hat{\gamma}^*$ w/o Replacement | Mean $\hat{\gamma}^*$ w/ Replacement | Mean $\gamma^*$ |
|---|---|---|---|
| 0.0 | 0.5595 | 0.3971 | 0.4788 |
| 0.2 | 0.6075 | 0.5762 | 0.5067 |
| 0.4 | 0.7390 | 0.7241 | 0.5542 |
| 0.6 | 0.8681 | 0.8198 | 0.6371 |
| 0.8 | 0.9019 | 0.8771 | 0.8014 |
| 1.0 | 1.0000 | 1.0000 | 1.0000 |

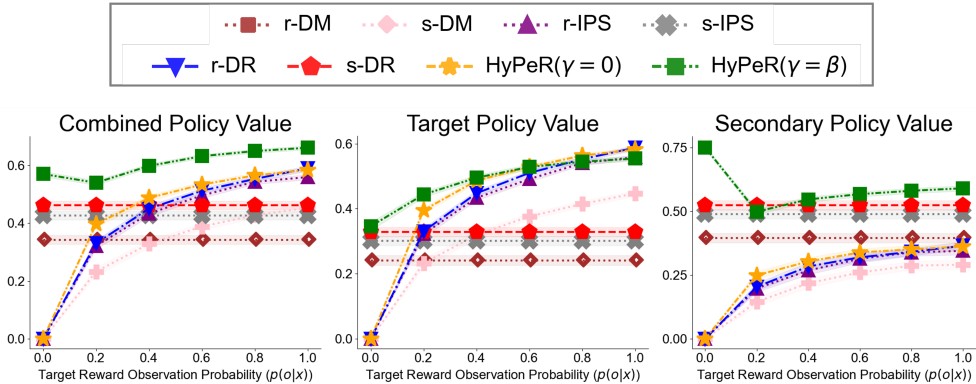

Figure 6: Comparing the combined, target, and secondary policy values of OPL methods with **varying target reward observation probabilities** ($p(o|x)$).

**Results.** Figures 6, 7, and 8 present additional experiments related to the research questions addressed in Section 5, including r-DM and s-DM. Specifically, Figure 6 evaluates the policy values under varying observation probabilities for the target rewards, Figure 7 evaluates the methods with different training data sizes, and Figure 8 presents results for different degrees of target-secondary reward correlation. Across all experimental scenarios, we observe that r-DM and s-DM perform significantly worse than the corresponding IPS and DR methods due to high bias in the regression estimates.

Figure 9 shows results with varying true weights $\beta$ in the combined objective. In this experiment, we additionally compare two more baseline methods: **HyPeR(Tuned $\hat{\gamma}^*$ w/o replacement)** and **DR with $F(s, r)$**. HyPeR(Tuned $\hat{\gamma}^*$ w/o replacement), unlike our tuning method, performs tuning of $\gamma$ using sampling **without replacement**, i.e, without bootstrapping. We compare this baseline method to empirically show the effectiveness of our tuning method of leveraging the bootstrapping procedure. DR with $F(s, r)$ is another additional baseline that naively combines the target and secondary rewards in its policy gradient estimation; it uses Doubly Robust where the reward is defined as $F(s, r) = r$ when $o = 1$, and otherwise $F(s, r) = F(s)$ as in the method s-DR.

In Figure 9, we also observe that the additionally compared baseline methods all performed worse than the corresponding DR methods. The result also demonstrates that parameter tuning *with* replacement empirically outperforms tuning *without* replacement for a range of the true weight $\beta$ values in the combined policy value objective. Moreover, by comparing HyPeR against the method of DR with $F(s, r)$, the results empirically demonstrate how combining the target and secondary rewards is indeed crucial and that HyPeR does it effectively.

We also share the tuned $\hat{\gamma}^*$ values and compare them with the ground truths in Table 2 below.

## C.2   ADDITIONAL SYNTHETIC EXPERIMENTS

In this section, we additionally address three more research questions: two questions regarding the optimization performance under varying noise levels in each reward type and one concerning the effectiveness of HyPeR with estimated observation probabilities $p(o|x)$.

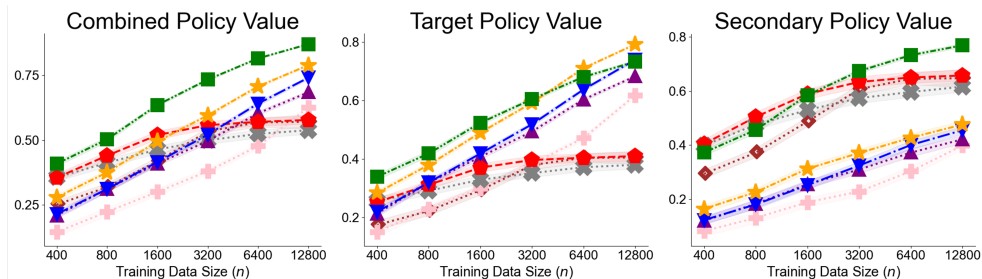

Figure 7: Comparing the combined, target, and secondary policy values of OPL methods with **varying training data sizes** ($n$)

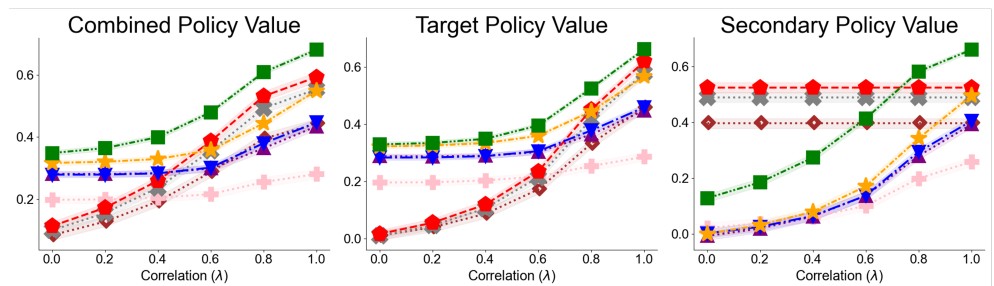

Figure 8: Comparing the combined, target, and secondary policy values of OPL methods with **varying degrees of target-secondary reward correlation** ($\lambda$). Secondary rewards can completely explain the target reward at $\lambda = 1$, and they are less correlated with smaller $\lambda$.

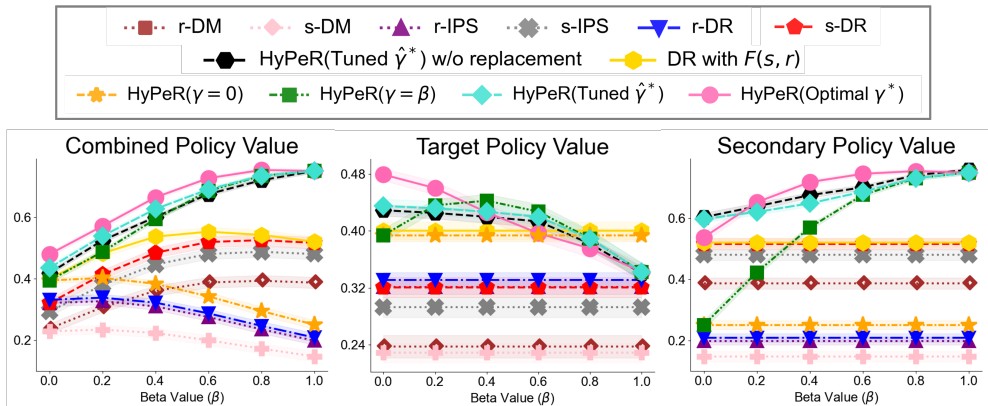

Figure 9: Comparing the combined, target, and secondary policy values of OPL methods with **varying $\beta$**, which is a weight that balances target and secondary policy values in the combined policy value. Higher $\beta$ means the secondary policy value becomes more dominant.

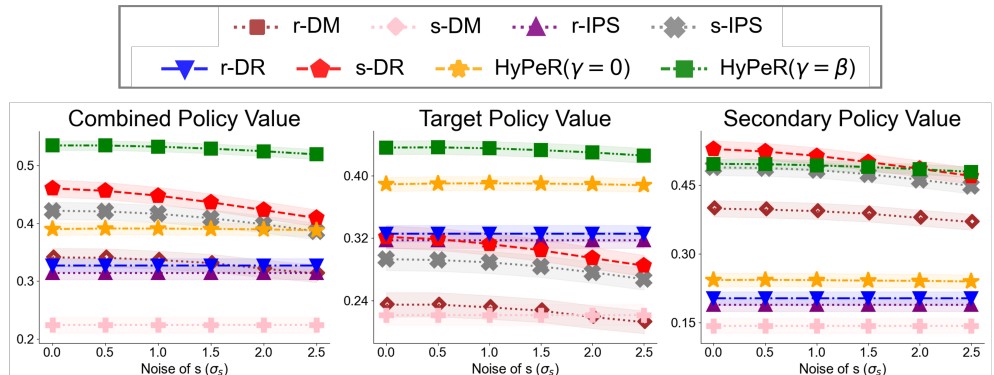

Figure 10: Comparing the combined, target, and secondary policy values of OPL methods with **varying noise of secondary reward** ($\sigma_s$).

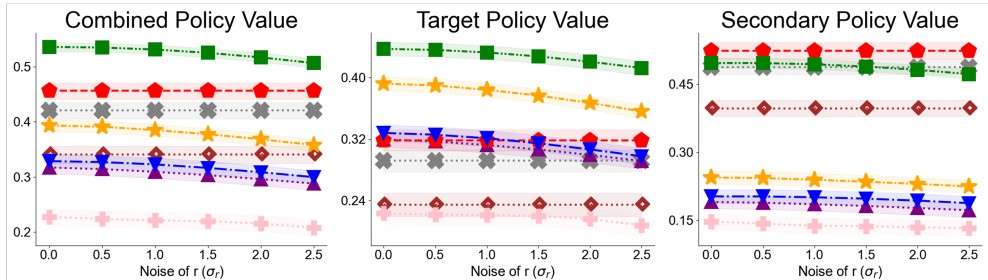

Figure 11: Comparing the combined, target, and secondary policy values of OPL methods with **varying noise of target reward** ($\sigma_r$).

**How does HyPeR perform under varying noise levels in secondary rewards?** Figure 10 evaluates the relative policy value when the noise level in the secondary rewards is varied. As the secondary rewards become noisier, their variance increases. Consequently, methods that incorporate secondary rewards in their estimation (i.e., HyPeR($\gamma = \beta$), HyPeR($\gamma = 0$), s-DR, s-IPS, and s-DM) perform worse as noise increases. Most notably, the performance of s-IPS and r-DR degrades the most. In contrast, while HyPeR-based methods are also affected by the noise, their performance does not degrade as severely as that of the baseline methods that rely solely on secondary rewards, demonstrating its robustness to the noise due to the effective combination of the two types of rewards.

**How does HyPeR perform under varying noise levels in target rewards?** Figure 11 presents the results for different noise levels in the target rewards. As the noise increases, the variance in the target rewards becomes higher. This affects methods that utilize target rewards in their estimation (i.e., HyPeR($\gamma = \beta$), HyPeR($\gamma = 0$), r-DR, r-IPS, and r-DM). While HyPeR methods are negatively impacted, they consistently outperform the baseline methods that rely exclusively on target rewards, showing robustness as well.

**How does HyPeR perform with estimated observation probabilities?** To investigate the robustness of HyPeR to estimated probabilities $p(o|x)$, we conducted experiments under unknown $p(o|x)$ with varying levels of noise on $o$. The results are shown in Figure 12, where the methods that rely on $p(o|x)$ are affected by the estimation error and the presence of noise. We observed, however, that HyPeR consistently achieves better combined policy values compared to the baseline methods even under the realistic case with estimated probabilities $p(o|x)$. In particular, HyPeR (Tuned $\hat{\gamma}^*$) demonstrates better robustness to the noise in $p(o|x)$, as it can adaptively assign greater weight (via the tuning procedure) to the secondary policy gradient when effective.

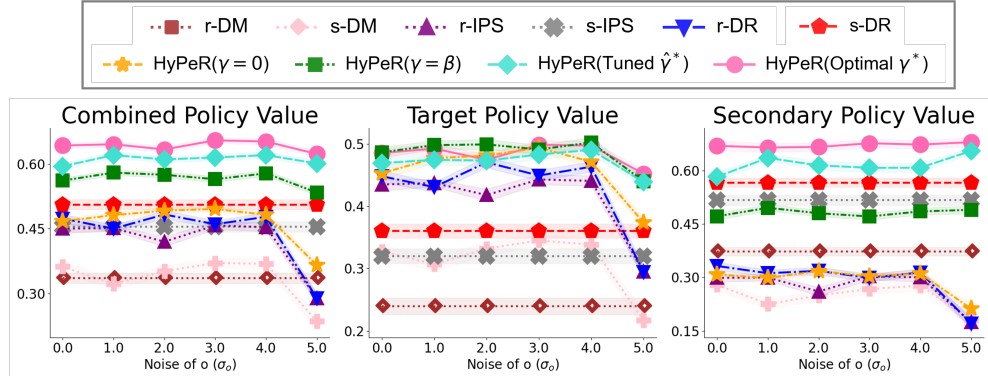

Figure 12: Comparing the combined, target, and secondary policy values of OPL methods with **varying noise in the observation probability** ($\sigma_o$). Note that the conditional probabilities ($p(o|x)$) are fully unknown and they are estimated based on the logged data.

Table 3: Comparison of $\gamma^*$ and $\hat{\gamma}^*$ values for different scenarios.

(a) Varying Target Reward Observation Probabilities ($p(o|x)$) (c.f. Figure 13)

| $p(o|x)$ | Mean $\gamma^*$ | Mean $\hat{\gamma}^*$ |
|---|---|---|
| 0.0 | 1.0000 | 1.0000 |
| 0.2 | 0.6262 | 0.5926 |
| 0.4 | 0.5811 | 0.5542 |
| 0.6 | 0.5124 | 0.5002 |
| 0.8 | 0.4771 | 0.4701 |
| 1.0 | 0.4299 | 0.4512 |

(b) Varying Training Data Sizes ($n$) (c.f. Figure 14)

| Data Size | Mean $\gamma^*$ | Mean $\hat{\gamma}^*$ |
|---|---|---|
| 500 | 0.7205 | 0.5458 |
| 1000 | 0.6612 | 0.5890 |
| 2000 | 0.5678 | 0.6392 |
| 4000 | 0.4462 | 0.5309 |
| 8000 | 0.3356 | 0.4794 |
| 16000 | 0.2878 | 0.4306 |

(c) Different Beta Values ($\beta$) (c.f. Figure 15)

| $\beta$ | Mean $\gamma^*$ | Mean $\hat{\gamma}^*$ |
|---|---|---|
| 0.0 | 0.5743 | 0.5287 |
| 0.2 | 0.6603 | 0.5833 |
| 0.4 | 0.7193 | 0.6386 |
| 0.6 | 0.7577 | 0.6549 |
| 0.8 | 0.8592 | 0.8021 |
| 1.0 | 1.0000 | 1.0000 |

(d) Different Action Sizes ($|\mathcal{A}|$) (c.f. Figure 16)

| Action Size | Mean $\gamma^*$ | Mean $\hat{\gamma}^*$ |
|---|---|---|
| 25 | 0.5998 | 0.5312 |
| 50 | 0.6057 | 0.5159 |
| 100 | 0.5686 | 0.5008 |
| 200 | 0.6241 | 0.4925 |
| 400 | 0.5972 | 0.5328 |
| 800 | 0.6124 | 0.5086 |

## C.3 DETAILED AND ADDITIONAL REAL-WORLD EXPERIMENT RESULTS

In this section, we share more detailed experiment results of the real-world data experiments in Section 6. We add r-DM, r-IPS, s-DM, and s-IPS as baselines compared to the main text.

**Results.** Figures 13, 14, and 15 present more detailed results from the real-data experiments in Section 6, including additional methods for comparison. Figure 13 evaluates the policy values under varying observation probabilities of the target rewards, while Figure 14 evaluates the methods with different training data sizes. Figure 15 evaluates the methods with different true weights $\beta$ in the objective. In this experiment, we also compare DR with $F(s, r)$ (the yellow line), which serves as a baseline that naively utilizes both target and secondary rewards when available.

In this section, we also evaluate the relative policy values when varying the size of the action set $|\mathcal{A}|$, as shown in Figure 16. As the action set size increases, the performance of all methods degrades due to an increase in variance in gradient estimation. However, we observe that HyPeR methods consistently outperform all other methods across all evaluated policy values.

Below, Tables 3, 3, 3, and 3 present the tuned $\hat{\gamma}^*$ values for the four real-world experiments, compared to the true optimal value $\gamma^*$, averaged over the number of simulations.

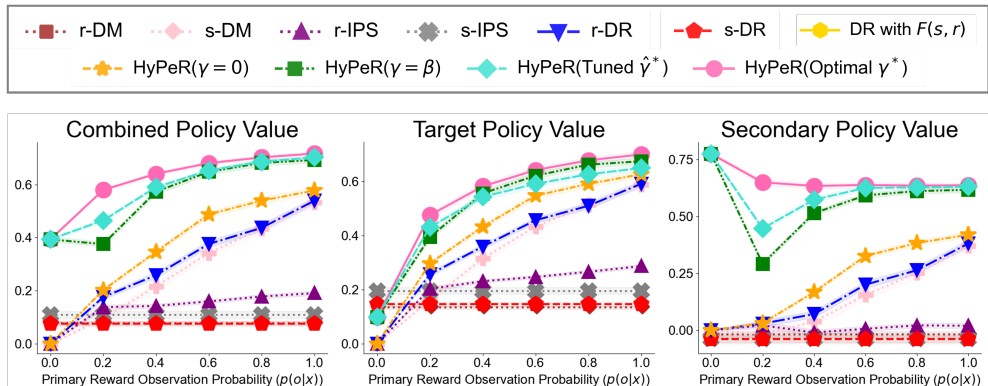

Figure 13: Comparing the combined, target, and secondary policy values of OPL methods with **varying target reward observation probabilities** ($p(o|x)$) on KuaiRec Dataset

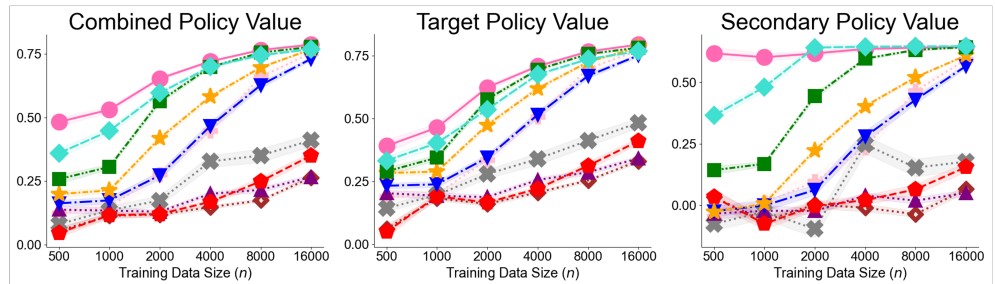

Figure 14: Comparing the combined, target, and secondary policy values of OPL methods with **varying training data sizes** ($n$) on KuaiRec Dataset

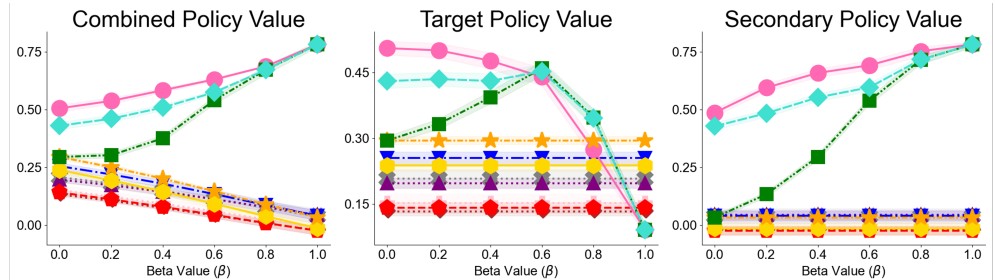

Figure 15: Comparing the combined, target, and secondary policy values of OPL methods with **varying** $\beta$, which is a weight that balances target and secondary policy values in the combined policy value. Higher $\beta$ means the secondary policy value becomes more dominant on KuaiRec Dataset.

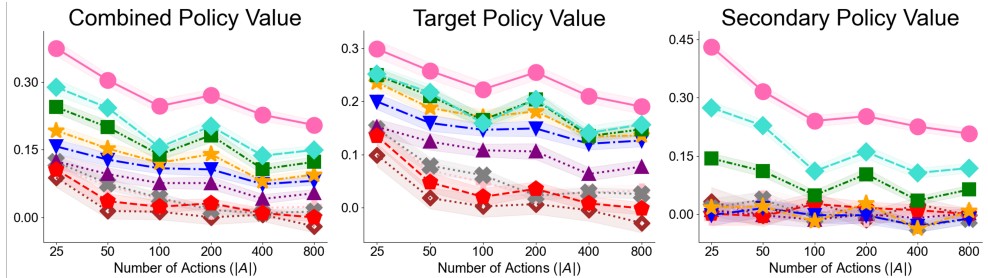

Figure 16: Comparing the combined, target, and secondary policy values of OPL methods with **varying action size** $|\mathcal{A}|$ on KuaiRec Dataset.

