# OpenReview forum: "A General Framework for Off-Policy Learning with Partially-Observed Reward"
_ICLR.cc/2025/Conference — ICLR 2025 Poster_

### Official Review · Reviewer_tksu · 2024-10-29

**Soundness:** 2
**Presentation:** 3
**Contribution:** 3
**Rating:** 6
**Confidence:** 4

**Summary:**

The paper deals with the problem of Off-Policy Learning (OPL) in the contextual bandit domain. They focus, in particular, on the case where a secondary reward is fully observed while the target reward is, in general, partially observed. This is the case in many real-world applications, such as recommendation systems. The paper introduces Hybrid Policy Optimization for partially observed Reward (HyPeR), a method for improving off-policy learning (OPL) when target rewards are partially observed. By utilizing secondary rewards, the method aims to reduce variance in policy gradient estimation, thus improving learning efficiency. An empirical analysis (on synthetic and real data) is presented to support theoretical findings.

**Strengths:**

- Addresses a relevant issue in OPL with partially observed rewards: the issue is very relevant, since there are many real-world application domains where the target reward may be partially observed, but there are a lot of secondary reward signals available. This paper makes good use of those signals and practitioners may be interested in this

- New estimator with theoretical properties: the HyPeR estimator for the policy gradient is still unbiased and provably reduces (as long as \hat{q}(x,a,s) is more accurate than \hat{q}(x,a)) the variance of the estimation by introducing a sort of "control variate" using the secondary rewards. This is an important contribution.

- Empirical analysis on two datasets: I like the fact that the empirical analysis is done on both synthetic and real data, and also with a lot of different settings (e.g., varying data size, secondary-target reward correlation, etc.).

**Weaknesses:**

- Section on the tuning procedure: in my opinion, this section may be a bit misleading (not intentionally, of course). This is because the authors claim that we can simply apply a cross-validation procedure to tune a hyperparameter as we do in the supervised setting. However, this is not so trivial as they claim: the fact that we are dealing with an Off-Policy problem means that the data is not iid with the target data. Hence, performing a cross-validation on such data will not give an unbiased estimation of the performance of the target policy. Nevertheless, it may be a heuristic way of performing hyperparameter tuning (I've seen papers in the OPE/OPL domain doing it), but it must be acknowledged that it is a biased way of performing hyperparameter tuning.

- I am not convinced of the design choices for the secondary targets in the experimental setting: let us consider the real data experiment. The authors say that the target reward is the watch ratio r. We would like however to provide a good PG estimate even when we do not observe this reward. They decide that they will do it with secondary rewards that they design to be *very* correlated with r. In particular, with s_1 and s_2, you can almost reconstruct r. Hence I am wondering if this experiment is really meaningful, since we would like to see what happens when we do not observe r, but we almost exactly observe it (albeit quantized: if s_2 = -1, then r < 0.5, if s_2 = 0 and s_1 = 0, then r \in [0.5, 2), if s_1=1, then r>= 2).

A very minor issue that did not impact the score:
- If I recall correctly, you never mention the fact that you focus only on contextual bandits in the introduction. Perhaps it is better to mention it somewhere in the intro, otherwise readers may be confused and think you are focusing on full RL.

**Questions:**

- Do you agree that the cross-validation procedure that you proposed is biased by design? In my opinion (if I am correct), you can still leave the section as is, but just acknowledge this caveat. Indeed, empirically, this procedure seems to work despite being biased, so it is not a big deal.

- Can you elaborate a bit on the design choices for the secondary rewards in your experimental settings?

- Can you provide ablation studies on the secondary reward dimensions? I would like to see which dimensions are responsible for the HyPeR good performance. My intuition would say that, in the real data case for example, it is because of s_1 and s_2, since you can almost construct the target reward r based on those.

- Can you provide an ablation study on what happens when you perform cross-validation with sampling without replacement? You claim that it is important to have sampling with replacement. Do you have empirical data to support this claim? It can be useful for researchers in the future


Finally, it is not a question but more of a comment:
Your estimator of the policy gradient also seems robust to missing secondary rewards: if you set \hat{q}(x,a,s) to resort to \hat{q}(x,a) when there are missing s (for any reason), your estimator reduces to Doubly Robust, which is still an unbiased estimator.

---

> ### Author Response · Authors · 2024-11-21
>
> We appreciate the valuable and thoughtful feedback from the reviewer. We respond to the concrete questions and comments in detail below.
>
> > Can you elaborate a bit on the design choices for the secondary rewards in your experimental settings?
>
> > Can you provide ablation studies on the secondary reward dimensions? I would like to see which dimensions are responsible for the HyPeR good performance. My intuition would say that, in the real data case for example, it is because of s_1 and s_2, since you can almost construct the target reward r based on those.
>
> Thank you for the question and the valuable suggestion. In the real-world experiment, we focused on designing secondary rewards that are indeed worth maximizing, as explained in Section 6. For example, $s_1$ aims to prioritize sessions with an exceptionally long watch ratio, and $s_2$ aims to raise the engagement floor. We agree with the reviewer that $s_1$ and $s_2$ are correlated with the target reward. However, it is worth noting that the other two secondary rewards do not necessarily correlate with the target reward, and our algorithm does not use any prior knowledge about the correlation structure.
>
>
> It is also interesting to see the center figure of Figure 14 in Appendix C.3. It shows that, in terms of the target policy value, the policy performance becomes much worse when we use only the secondary rewards (i.e., HyPeR’s performance at $\beta=1.0$), compared to the case where we use the only target reward (i.e., performance of r-DR). Similarly, the right-hand figure in Figure 14 demonstrates that the maximization of the target reward (i.e., performance of r-DR) does not improve the secondary policy value at all. This demonstrates that the sum of the secondary rewards that we used in our real experiment is only slightly correlated with the target reward. We will mention this observation in the main text in the revision.
>
>
> > Section on the tuning procedure: in my opinion, this section may be a bit misleading (not intentionally, of course). This is because the authors claim that we can simply apply a cross-validation procedure to tune a hyperparameter as we do in the supervised setting. However, this is not so trivial as they claim: the fact that we are dealing with an Off-Policy problem means that the data is not iid with the target data. Hence, performing a cross-validation on such data will not give an unbiased estimation of the performance of the target policy. Nevertheless, it may be a heuristic way of performing hyperparameter tuning (I've seen papers in the OPE/OPL domain doing it), but it must be acknowledged that it is a biased way of performing hyperparameter tuning.
>
> > Do you agree that the cross-validation procedure that you proposed is biased by design? In my opinion (if I am correct), you can still leave the section as is, but just acknowledge this caveat. Indeed, empirically, this procedure seems to work despite being biased, so it is not a big deal.
>
> Thank you for the interesting questions. First, we would like to point out a potential critical misunderstanding made by the reviewer. That is, our data-generating process is indeed i.i.d (same as most other OPE papers), as formulated in Eq.(6), and thus there is no bias due to the data being non-i.i.d.
>
> > Can you provide an ablation study on what happens when you perform cross-validation with sampling without replacement? You claim that it is important to have sampling with replacement. Do you have empirical data to support this claim? It can be useful for researchers in the future.
>
> Thank you for the interesting suggestion. Initially, we implemented the tuning procedure without replacement but found that it did not perform well. The issue is that the training data size during cross-validation becomes significantly smaller than that used in the actual training, making it difficult to precisely evaluate the variance of the policy gradient estimator and resulting in ineffective tuning. That is how we arrived at the current proposed procedure with replacement to ensure that the training data size during tuning is the same as that during actual learning.
>
>
> The following table reports our additional experiment results comparing our method with and without replacement during weight tuning on synthetic data. The result suggests that tuning with replacement empirically outperforms tuning without replacement for a range of the true weight $\beta$.
>
>
> | True weight $\beta$                | 0.0 | 0.2 | 0.4 | 0.6 | 0.8 | 1.0 |
> |-----------------------|----------|----------|----------|----------|----------|----------|
> | HyPeR (Tuned $\hat{\gamma}^*$ w/o replacement) | 0.4181 | 0.5238 | 0.6000 | 0.6741 | 0.7196 | 0.7497 |
> | HyPeR (Tuned $\hat{\gamma}^*$ w/ replacement)   | 0.4350 | 0.5382 | 0.6261 | 0.6898 | 0.7339 | 0.7497 |
> | HyPeR (Optimal $\gamma^*$)                          | 0.4792 | 0.5707 | 0.6639 | 0.7261 | 0.7527 | 0.7497 |

---

> > ### Comment · Reviewer_tksu · 2024-11-22
> > **Response**
> >
> > Thanks for the detailed response.
> >
> > I think everything you said makes sense. However, I want to clarify regarding the validation procedure.
> >
> > I know that the data is still iid, but what I wanted to say is that it is not iid with the **target** data distribution.
> >
> > Mathematically speaking, let us say that we have an estimator $\hat{V}$ that wants to estimate the policy value $V$.
> > Let the contexts, logged actions, rewards be distributed as $x \sim p(\cdot), a \sim \pi_0(\cdot|x), r \sim p(\cdot|x, a)$, where $\pi_0$ is the logging distribution. Let us call this data distribution $P_{log}$. What we want to estimate is, given a new (evaluation) policy $\pi_e$, what is the expected reward this policy would obtain:
> >
> > $V(\pi_e) = E_{P_{target}}[r]$
> >
> > where $P_{target} =  p(\cdot)  \pi_e(\cdot|x) p(\cdot|x, a)$ (notice the distribution shift due to $\pi_e \neq \pi_0$).
> >
> > Now, if our estimator is parametric: $\hat{V}(\pi_e; \gamma)$, what I would expect is to tune the parameter in order to estimate in the best way possible the real policy, for instance minimizing the MSE:
> >
> > $\gamma_1 = \arg \min_\gamma (V(\pi_e) - \hat{V}(\pi_e; \gamma))^2$,
> >
> > which is of course unfeasible in this naive way because we do not have access to $V(\pi_e)$. (Notice that there are papers [1] that showed alternative ways to perform this minimization, which may not apply to your scenario, though)
> >
> > Then, **after this**, you would perform OPL maximizing the estimator $\hat{V}(\pi_e; \gamma_1)$.
> >
> > However, what you do is to assume already that your estimator is perfectly aligned with $V$ for all possible values of $\gamma$. Hence, a maximization of the estimated $\hat{V}$ would imply a maximization of the expected $V$ (which is what we really care about). Therefore, you try to find:
> >
> > $\gamma^* = \arg \max_\gamma \hat{V}(\pi_e; \gamma)$
> >
> > This strong assumption may have some drawbacks, which is exactly the point of [2].
> > If you agree, I think you should definitely acknowledge [2] and the possible drawbacks of this tuning, without necessarily change anything in the procedure.
> >
> > [1]: Su, Y.; Srinath, P.; and Krishnamurthy, A. 2020. Adaptive Estimator Selection for Off-Policy Evaluation.
> >
> > [2]: Saito and Nomura, 2024. Hyperparameter Optimization Can Even be Harmful in Off-Policy Learning and How to Deal with It.

---

> > > ### Author Response · Authors · 2024-11-22
> > >
> > > We would like to thank the reviewer for clarifying their initial review. We believe we now fully understand the reviewer’s argument regarding the tuning procedure.
> > >
> > > We completely agree with the reviewer’s point on hyperparameter tuning, particularly in the context of OPL. We are also familiar with [1] and [2] (as well as a few other relevant papers such as [3, 4]). Regarding [1], we think that the reviewer is referring to their monotonicity assumption, which may not hold in our problem of tuning $\gamma$. Furthermore, while we are mainly discussing a tuning procedure for OPL, [1] focuses on the estimator selection problem for OPE. Nonetheless, it is indeed a relevant work, and we will touch on it in our text.
> > >
> > > We also agree with the reviewer’s observation that [2] is more relevant, as it focuses on the potential failures of a typical or naive tuning procedure, such as using standard estimators like IPS and DR to estimate the validation performance for hyperparameter tuning in OPL. The scenario studied in [2] can indeed apply to our tuning procedure, and it is possible that incorporating the correction methods from [2] could further improve our method by yielding a more effective $\gamma$.
> > > **However, it is also worth noting that, in our experiments, the parameter tuned via the standard tuning procedure $\hat{\gamma}$ is already nearly optimal compared to the true optimal parameter $\gamma^{\*}$ regarding the combined policy value (as shown in Figures 4 and 5).**
> > >
> > > We have mentioned [2], revised the relevant sections of the text, and updated the paper to reflect the reviewer’s valuable suggestion. (Additionally, we have addressed all the typos kindly pointed out by other reviewers, which are now highlighted in red in our text.)
> > >
> > > **We would like to confirm whether this addresses the reviewer’s main concern. If there are any additional concerns that might prevent the reviewer from reconsidering their score, we would be more than happy to discuss further.**
> > >
> > > ---
> > > [3] Nicolò Felicioni, Michael Benigni, Maurizio Ferrari Dacrema. Automated Off-Policy Estimator Selection via Supervised Learning https://arxiv.org/abs/2406.18022
> > >
> > > [4] Takuma Udagawa, Haruka Kiyohara, Yusuke Narita, Yuta Saito, and Kei Tateno. Policy-adaptive estimator selection for off-policy evaluation. AAAI2023.

---

> > > > ### Comment · Reviewer_tksu · 2024-11-22
> > > > **Thank you**
> > > >
> > > > Thanks for uploading the revised version.
> > > >
> > > > I have updated my score.

---

> > > > > ### Author Response · Authors · 2024-11-24
> > > > >
> > > > > We greatly appreciate the timely and thoughtful discussion provided by the reviewer once again.
> > > > > For the reviewer’s information, we have updated our draft and included all the additional experiments suggested by the reviewer and other reviewers. More specifically:
> > > > >
> > > > > - We mentioned the need for estimating the observation probabilities $p(o|x)$ in Section 4 and added experimental results with estimated probabilities $p(o|x)$ in the appendix.
> > > > > - We added a baseline of using $F(s, r) = o \cdot r + (1 - o) \cdot F(s)$ in the appendix.
> > > > > - We clarified in Section 6 that, in the real-world experiment, the target and secondary rewards are not overly correlated, making it a non-trivial problem and crucial to effectively combine them, based on empirical evidence in the appendix.
> > > > > - We empirically demonstrated the advantage of performing sampling with replacement (compared to sampling without replacement) when tuning the key parameter $\gamma$ in our method in the appendix.
> > > > > - We mentioned the caveat of parameter tuning for OPL, as recently studied in (Saito and Nomura 2024).
> > > > > - We more explicitly state that our focus is on the contextual bandit setup, and that we leave a more general RL setup for future research.
> > > > >
> > > > >
> > > > > In our understanding, we have now addressed all the weaknesses and questions raised by the reviewers. We do not intend to challenge the reviewer’s evaluation, however, we would like to ask what prevents the reviewer from giving a clear acceptance for now, if possible. **Our main intention is to seek feedback and to understand the background of the evaluation so that we can further improve not only this submission but also our future research practices. We would also be more than happy to discuss any remaining questions regarding our work.**

---

> > > > > > ### Author Response · Authors · 2024-11-30
> > > > > >
> > > > > > Thank you so much once again for the reviewer’s effort in evaluating our work. We would now like to address the following comments from the initial review more rigorously.
> > > > > >
> > > > > >
> > > > > > >I am not convinced of the design choices for the secondary targets in the experimental setting... ... In particular, with s_1 and s_2, you can almost reconstruct r. Hence I am wondering if this experiment is really meaningful, since we would like to see what happens when we do not observe r, but we almost exactly observe it
> > > > > >
> > > > > > > Can you provide ablation studies on the secondary reward dimensions? I would like to see which dimensions are responsible for the HyPeR good performance. My intuition would say that, in the real data case for example, it is because of s_1 and s_2, since you can almost construct the target reward r based on those.
> > > > > >
> > > > > > Specifically, we conducted two additional ablation studies on the real-world dataset to address the reviewer’s comment regarding the secondary rewards.
> > > > > >
> > > > > > First, we present experimental results with various combinations of secondary reward dimensions in the following, focusing particularly on the first two dimensions, $s_1$​ and $s_2$​, which are relatively highly correlated with the target reward. The results indicate that, in terms of both the target and combined policy values, **HyPeR remains the best option even when either $s_1$​ or $s_2$​, or both, are unobserved.** The results also show that incorporating $s_1$ and/or $s_2$ improves HyPeR’s performance, which validates the reviewer’s initial observation. However, **the additional results clearly demonstrate that HyPeR substantially outperforms the baseline methods for any potential construction of $s$.**
> > > > > >
> > > > > > "**Combined Policy Value**"
> > > > > >
> > > > > > | Used dimension of $s$ | $\\{s_1, s_2, s_3, s_4\\}$  | $\\{s_1, s_3, s_4\\}$ | $\\{s_2, s_3, s_4\\}$ | $\\{s_3, s_4\\}$|
> > > > > > |-------------------|----------|----------|----------|----------|
> > > > > > | r-DR      | 0.2025   | 0.1982   | 0.2272   | 0.1920   |
> > > > > > | s-DR      | 0.1198   | -0.2381  | 0.2048   | -0.4214  |
> > > > > > | HyPeR($\gamma=0$)         | 0.2368    | 0.2259  | 0.2534  | 0.2339   |
> > > > > > | **HyPeR($\gamma=\beta$)**    | **0.3297**   | **0.3222** | **0.2738**  | **0.2556**   |
> > > > > >
> > > > > > "**Target Policy Value**"
> > > > > >
> > > > > > | Used dimension of $s$  | $\\{s_1, s_2, s_3, s_4\\}$  | $\\{s_2, s_3, s_4\\}$  | $\\{s_1, s_3, s_4\\}$ | $\\{s_3, s_4\\}$|
> > > > > > |-------------------|----------|----------|----------|----------|
> > > > > > | r-DR      | 0.2618   | 0.2618   | 0.2618 | 0.2618   |
> > > > > > | s-DR      | 0.1662  | -0.0251 | -0.0551 | -0.6050  |
> > > > > > | HyPeR($\gamma=0$)         | 0.3037   | 0.2916  | **0.2964**  | **0.2685**  |
> > > > > > | **HyPeR($\gamma=\beta$)**    | **0.3598**   | **0.3471** | 0.2743  | 0.2493   |
> > > > > >
> > > > > >
> > > > > > Additionally, we analyzed the results under varying noise levels in $s_1$​ and $s_2$​ to investigate the effect of the quality of $s$ on HyPeR’s performance on real-world data. From these results, we observe that HyPeR outperforms the baseline methods across all tested noise levels. Although the performance difference is largest when there is no noise ($\sigma_s=0$), this additional ablation demonstrates that **HyPeR remains the best method even when the secondary rewards are highly noisy** in our real-world experiments.
> > > > > >
> > > > > >
> > > > > > "**Combined Policy Value**"
> > > > > >
> > > > > > | Noise on $s$ ($\sigma_s$)            | 0.0 | 0.5 | 1.0 | 1.5 | 2.0 | 2.5 |
> > > > > > |-------------------|----------|----------|----------|----------|----------|----------|
> > > > > > | r-DR      | 0.1869   | 0.1904   | 0.1889   | 0.1863   | 0.1860   | 0.1842   |
> > > > > > | s-DR      | 0.1402   | 0.1854   | 0.0093   | 0.0345   | 0.0688   | 0.0876   |
> > > > > > | HyPeR($\gamma=0$)        | 0.2944   | 0.2023   | 0.1835   | 0.1870   | 0.2186   | 0.2358   |
> > > > > > | **HyPeR($\gamma=\beta$) **   | **0.3793**   | **0.2555**   | **0.2611**   |  **0.1914**   | **0.2695**   | **0.2760**   |
> > > > > >
> > > > > >
> > > > > > "**Target Policy Value**"
> > > > > >
> > > > > > | Noise on $s$ ($\sigma_s$)            | 0.0 | 0.5 | 1.0 | 1.5 | 2.0 | 2.5 |
> > > > > > |-------------------|----------|----------|----------|----------|----------|----------|
> > > > > > | r-DR    | 0.2483   | 0.2483   | 0.2483   | 0.2483   | 0.2483   | 0.2483   |
> > > > > > | s-DR     | 0.1998   | 0.1948   | 0.0696   | 0.0642   | 0.0917   | 0.1104   |
> > > > > > | HyPeR($\gamma=0$)          | 0.3332   | 0.2512   | 0.2533   | 0.2546   | 0.2835   | **0.3123**   |
> > > > > > |HyPeR($\gamma=\beta$)    | **0.3870**   | **0.2805**   | **0.2964**   | **0.2561**   | **0.2923**   | 0.2984   |
> > > > > >
> > > > > > We have included these interesting additional observations in the appendix of the paper. Once again, we deeply appreciate the reviewer’s thoughtful comments and suggestions, which have significantly improved our work. If the reviewer has any further questions, comments, or feedback on our work, we would be delighted to address them.

---

### Official Review · Reviewer_8JAf · 2024-11-04

**Soundness:** 2
**Presentation:** 2
**Contribution:** 3
**Rating:** 6
**Confidence:** 4

**Summary:**

The paper addresses the problem of off-policy learning in contextual bandits where rewards are only partially observable. In this setting, the authors propose using an auxiliary variable to reduce the variance of the double-robust inverse propensity weighted (IPW) estimator for the reward. They introduce a Hybrid Policy Optimization for Partially-Observed Reward (HyPeR) estimator, which incorporates an auxiliary-conditioned reward estimator to refine average reward predictions. Theoretical and empirical results demonstrate that when the auxiliary variable is highly correlated with the target reward, the HyPeR method surpasses baseline approaches in performance.

**Strengths:**

Incorporating auxiliary variables for variance reduction is a well-established idea in the ML community. However, the authors innovatively integrate this concept with the doubly robust estimator to tackle the challenge of limited data coverage under partially observable rewards—a contribution that appears novel. The experiments further support the theoretical insights, and the paper is generally clear and easy to follow, aside from a few errors and typos.

**Weaknesses:**

I noticed several potential technical issues (or possibly typos) that may impact the rigor of the results:

1. In eqn (6), you write down the distribution in a way that allow $r_i$ to also depend on $o_i$. This will cause the problem of confounding bias, which does not give rise to an unbiased estimator of $q(x, a)$. See Kato et al. 2021 or Pearl, 2009 for details. Is this a typo or is there something missing? I found in the proof of Theorem 1 that you indeed assume $r$ to be independent of $o$ when conditioned on other observable variables $(x, a, s)$. If this is an assumption, you should clearly state it.

2. In Eqn (10), you perhaps miss out a $o_i / p(o_i\,|\, x_i)$ term in the second line, as you cannot evaluate this term when $o_i = 0$ and $r_i$ is missing from the observational data.

3. In the proof of Theorem 2, what are $s_\theta^{(j)}(x, a)$, $\sigma^2(x, a, s, o)$? Are they also typos? You also miss out a variance symbol in the first line of the proof.


Despite these concerns, I believe the variance reduction claim remains valid, as the paper’s approach can be interpreted as an application of the well-known Rao-Blackwell theorem.

***References***
1. Kato, M., Imaizumi, M., McAlinn, K., Kakehi, H., & Yasui, S. (2021). Learning causal models from conditional moment restrictions by importance weighting. arXiv preprint arXiv:2108.01312.
2. Pearl, J. (2009). Causality. Cambridge university press.

**Questions:**

No further questions. My score reflects my current concerns about the technical soundness of this paper. I'm open to further discussions.

**Details Of Ethics Concerns:**

No concerns

---

> ### Author Response · Authors · 2024-11-21
>
> We appreciate the valuable and thoughtful feedback from the reviewer. We respond to the concrete questions and comments in detail below.
>
> > In eqn (6), you write down the distribution in a way that allow $r_i$ to also depend on $o_i$. This will cause the problem of confounding bias, which does not give rise to an unbiased estimator of $q(x,a)$. See Kato et al. 2021 or Pearl, 2009 for details. Is this a typo or is there something missing? I found in the proof of Theorem 1 that you indeed assume $r$ to be independent of $o$ when conditioned on other observable variables
> (x, a, s). If this is an assumption, you should clearly state it.
>
> This is a good catch. This is indeed a typo.
> We meant to write $\mathcal{D} := \\{ (x\_i, a\_i, o\_i, s\_i, r\_i) \\}\_{i=1}^n \sim p(\mathcal{D}) = \prod\_{i=1}^n p(x\_i) \pi\_0(a\_i \mid x\_i) p(o\_i \mid x\_i) p(s\_i \mid x\_i, a\_i) p(r\_i \mid x\_i, a\_i, s\_i)$,
>
> where $r$ is independent from $o$. We have fixed this.
>
> > In Eqn (10), you perhaps miss out a $o_i/p(o_i|x_i)$ term in the second line, as you cannot evaluate this term when $o_i=0$ and $r_i$ is missing from the observational data.
>
> Thank you for catching this too. This was indeed not intentional, and we do need $o_i/p(o_i|x_i)$ in the second line. We have already fixed this as well.
>
> > In the proof of Theorem 2, what are $s_i^{(j)}(x, a)$, $\sigma^2(x, a, s, o)$? Are they also typos? You also miss out a variance symbol in the first line of the proof.
>
> Thank you for pointing these out. $s_i^{(j)}(x, a)$ was meant to be $g_i^{(j)}(x, a)$, which is the policy score function of the $j$-th element. In addition, we meant to write $\sigma^2(x, a, s)$, which is equal to $\mathbb{V}[r|x, a, s]$, instead of $\sigma^2(x, a, s, o)$. We have fixed all of these and added the variance symbol already, and they do not affect our main results.
>
> If the reviewer has any technical points to discuss that may currently prevent them from reconsidering the score, we would be more than happy to discuss further.

---

> > ### Comment · Reviewer_8JAf · 2024-11-22
> >
> > Thanks for carefully addressing my concerns. I don't have further questions.

---

### Official Review · Reviewer_G83n · 2024-11-04

**Soundness:** 3
**Presentation:** 3
**Contribution:** 3
**Rating:** 6
**Confidence:** 4

**Summary:**

This work presents an off-policy learning approach to efficiently optimise partially-observed reward. The authors first model the data generation process of this problem, assuming access to dense, secondary rewards that are correlated to the target reward. They introduce HyPer, an unbiased policy gradient method that leverage secondary rewards to better optimise the true target. Experiments validate the approach on synthetic and semi-synthetic problems.

**Strengths:**

- The paper is well written and is easy to follow.
- The problem of partially observed/delayed reward is of high importance.
- The formulation of the data-generation process is simple and can model diverse problems.

**Weaknesses:**

- The contribution of the paper is twofold, first, the introduction of the framework of partially-observed rewards with presence of secondary rewards, that comes with a new data-generation-process (DGP), and secondly, the derivation of an unbiased policy gradient that can efficiently leverage all information to optimise the target. These two main contributions present some weaknesses:
   + The framework's data generation process (DGP) is **restrictive and was not properly tested**: If the DGP models missing, non delayed rewards, the claim that it can model delayed rewards is not supported, as these rewards can be observed after many interactions with the system and attribution of the reward to a certain action becomes the main difficulty [1].
   + The policy gradient approach relies on $p(o|x)$ **which is unknown**, thus the PG is never unbiased as we need to model $p(o|x)$. In addition, this quantity can be really hard to model in real life scenarios (For example, we do not know what explains why a user did not purchase an item after he clicked on it).
- The introduction of the combined policy value overloads the paper, and the story would have been simpler by defining a clear goal, which is to optimise the target reward, considered the north start in the majority of applications.
- There is a problem in Equation (10), which is the core result of the paper. The gradient boils down to a doubly robust gradient that does not use any secondary reward. This is a major typo that needs fixing.
- The paper does not discuss important, theoretical contributions to OPL, even in the extended related work. These contributions are based on the pessimism principle and were proven to be optimal, contrary to using naive policy gradients. The following lines of work should be incorporated to the extended related work and discussed:
   + Counterfactual Risk Minimization: Learning from Logged Bandit Feedback. Adith Swaminathan, Thorsten Joachims
   + Bayesian Counterfactual Risk Minimization. Ben London, Ted Sandler Proceedings of the 36th International Conference on Machine Learning, PMLR 97:4125-4133, 2019.
   + PAC-Bayesian Offline Contextual Bandits With Guarantees. Otmane Sakhi, Pierre Alquier, Nicolas Chopin Proceedings of the 40th International Conference on Machine Learning, PMLR 202:29777-29799, 2023.
   + Importance-Weighted Offline Learning Done Right. Germano Gabbianelli, Gergely Neu, Matteo Papini Proceedings of The 35th International Conference on Algorithmic Learning Theory, PMLR 237:614-634, 2024.
   + Logarithmic Smoothing for Pessimistic Off-Policy Evaluation, Selection and Learning. Otmane Sakhi, Imad Aouali, Pierre Alquier, Nicolas Chopin.

[1] Fixed point label attribution for real-time bidding. Bompaire et. al.

**Questions:**

In addition to the weaknesses, I have the following questions:
- How can we test if the DGP hold in real world problems? The real world experiment section in your paper is still a synthetic experiment simulated (with the DGP you suggest) on real world data.
- In real world scenarios, we never have access to the observation probability $p(o|x)$. How do you estimate it in the real world? And what is the impact of its estimation in the efficiency of your algorithm?
- Why did you need to introduce the combined policy value? And why did you define it with the sum of secondary rewards? In addition, It is better to change the name to "policy combined value", "policy secondary value" as it reflects more the quantities defined.
- The baselines s-IPS and s-DR use aggregation function $F(s)$ over the secondary rewards, why they do not use also the primary reward when available, giving $F(s, r)$?
- If it is difficult to derive the theoretical value of $\gamma^*$ as defined in your paper, one can derive the theoretical value of $\gamma^{**}$ to minimise the MSE of the gradient, did you explore this path?
- There is a new line of work [2] that optimises for long term reward, using dense, secondary/surrogate reward, is there a reason why you did not compare to it?
- Is there a reason why you did not include the large spectrum of pessimistic OPL?

[2] Long-term Off-Policy Evaluation and Learning. Yuta Saito, Himan Abdollahpouri, Jesse Anderton, Ben Carterette, Mounia Lalmas

---

> ### Author Response · Authors · 2024-11-21
>
> We appreciate the valuable and thoughtful feedback from the reviewer. We respond to the concrete questions and comments in detail below.
>
> > The framework's data generation process (DGP) is restrictive and was not properly tested: If the DGP models missing, non delayed rewards, the claim that it can model delayed rewards is not supported, as these rewards can be observed after many interactions with the system and attribution of the reward to a certain action becomes the main difficulty [1].
> How can we test if the DGP hold in real world problems? The real world experiment section in your paper is still a synthetic experiment simulated (with the DGP you suggest) on real world data.
>
> Thank you for raising the good points. **We argue that all the partial observation scenarios discussed in Section 1 can be addressed by our formulation and method.** In particular, we would like to highlight a critical misunderstanding by the reviewer regarding the delayed reward setup. In this problem, $r$ represents the *observed* reward, and if it is missing, we define it as $r = N/A$, as described around L164. The observation indicator $o$ specifies whether the reward is observed ($o=1$) or unobserved due to delay ($o=0$). This formulation is consistent with the real-world delayed reward setup. Therefore, it is unnecessary to verify whether this DGP is reasonable when it is already known that the reward is delayed.
>
>
> > The policy gradient approach relies on $p(o|x)$ which is unknown, thus the PG is never unbiased as we need to model $p(o|x)$. In addition, this quantity can be really hard to model in real life scenarios (For example, we do not know what explains why a user did not purchase an item after he clicked on it).
>
> > In real world scenarios, we never have access to the observation probability $p(o|x)$. How do you estimate it in the real world? And what is the impact of its estimation in the efficiency of your algorithm?
>
> This is a good question, and we are happy to clarify. We can estimate probabilities by solving a classification task that predicts $o$ from $x$ based on the available data. The reviewer particularly highlighted the difficulty of feature selection in estimating such probabilities. However, this challenge applies not only to the estimation of $p(o|x)$ but also to other aspects of the DGP, such as $p(r|x)$, which every method, including the baselines, must address. In practice, we can perform feature selection in a data-driven manner using cross-validation based on the policy value, as we always do.
>
>
> Additionally, to investigate the robustness of our method to estimated probabilities $p(o|x)$, we performed additional experiments under unknown $p(o|x)$ and with varying noise. The results are summarized in the following tables. We can see that HyPeR is yet the most effective method despite the need to estimate $p(o|x)$. The results also demonstrate that HyPeR(Tuned $\gamma^*$) is particularly robust to the noise in $p(o|x)$, as it can adaptively add more weight to the secondary policy gradient when effective. We will add these interesting observations in the revision and thank the reviewer for guiding us to these results.
>
>
> **Combined Policy Value**
> | Noise of $o$ ($\sigma_o$)  | 0.0 | 1.0 | 2.0 | 3.0 | 4.0 | 5.0 |
> |-------------------|----------|----------|----------|----------|----------|----------|
> | r-DR       | 0.4616   | 0.4792   | 0.4707   | 0.4641   | 0.4605   | 0.3012 |
> | s-DR      | 0.5193   | 0.5193   | 0.5193   | 0.5193   | 0.5193   | 0.5193   |
> | HyPeR($\gamma=\beta$)     | 0.5811   | 0.5850   | 0.5686   | 0.5839   | 0.5728   | 0.5461 |
> | HyPeR(Tuned $\hat{\gamma}^*$)  | 0.6223   | 0.6167   | 0.6186   | 0.6245   | 0.6054   | 0.6076 |
> | HyPeR(Optimal $\gamma^*$)  | 0.6479   | 0.6394   | 0.6541   | 0.6558   | 0.6469   | 0.6290 |
>
> **Target Policy Value**
> | Noise of $o$ ($\sigma_o$)  | 0.0 | 1.0 | 2.0 | 3.0 | 4.0 | 5.0 |
> |-------------------|----------|----------|----------|----------|----------|----------|
> | r-DR     | 0.4559   | 0.4546   | 0.4586   | 0.4399   | 0.4233   | 0.3048  |
> | s-DR     | 0.3810   | 0.3810   | 0.3810   | 0.3810   | 0.3810   | 0.3810  |
> | HyPeR($\gamma=\beta$)    | 0.5052   | 0.5104   | 0.4957   | 0.5120   | 0.5006   | 0.4573 |
> | HyPeR(Tuned $\hat{\gamma}^*$)  | 0.4849   | 0.4853   | 0.4874   | 0.5007   | 0.4798  | 0.4505   |
> | HyPeR(Optimal $\gamma^*$)    | 0.4989   | 0.4853   | 0.5014   | 0.5034   | 0.4941  | 0.4619  |

---

> ### Author Response · Authors · 2024-11-21
> **Official Comment by Authors (cont'd)**
>
> > Why did you need to introduce the combined policy value? And why did you define it with the sum of secondary rewards? In addition, It is better to change the name to "policy combined value", "policy secondary value" as it reflects more the quantities defined.
>
> Thank you for the questions and valuable suggestions. As we stated in the introduction and formulation sections, we introduced the combined policy value because there exist various real-life scenarios where we aim to optimize both the target reward and the secondary reward. We incorporate such a prevalent motivation by defining the weighted sum of the two reward types as the combined policy value. This is the most simple, interpretable, and common way for controlling the prioritization [1] [2].
>
>
> [1] Imo3:Interactive multi-objective off-policy optimization. Nan Wang, Hongning Wang, Maryam Karimzadehgan, Branislav Kveton, and Craig Boutilier.
>
> [2] Pessimistic off-policy multi-objective optimization. Shima Alizadeh, Aniruddha Bhargava, Karthick Gopalswamy, Lalit Jain, Branislav Kveton, and Ge Liu
>
> > The baselines s-IPS and s-DR use aggregation function $F(s)$ over the secondary rewards, why they do not use also the primary reward when available, giving F(s, r)?
>
> The s-DR and s-IPS estimators are only there as a baseline for the case where one type of reward was used because previously, it was common to rely either on dense secondary rewards or sparse target rewards for policy learning. The reviewer’s example ‘using target reward $r$ when available’ would not work well unless there is a suitable function $F$, which is _unknown_. In fact, finding a suitable function that appropriately balances the two rewards in a data-driven fashion is fundamentally what we were doing in our proposed tuning procedure, and this is not an already existing baseline.
>
> > If it is difficult to derive the theoretical value of $\\gamma^*$ as defined in your paper, one can derive the theoretical value of $\\gamma^{**}$ to minimize the MSE of the gradient, did you explore this path?
>
> Yes, we did think about it. However, there is no guarantee that the minimization of MSE leads to a better performance in OPL. MSE is often used in OPE only because minimizing it is the ultimate goal of the OPE problem. In OPL, the ultimate goal is to maximize the test policy value, and thus it is reasonable to perform cross-validation regarding that metric rather than MSE.
>
> > There is a new line of work [2] that optimises for long term reward, using dense, secondary/surrogate reward, is there a reason why you did not compare to it?
>
> Yes, we are aware of the work by Saito et al. However, as we have already discussed in the related work section, LOPE is critically different from our work. Specifically, LOPE considers the historical data of the form $\\mathcal{D}= \\{(x_i, a_i, s_i, r_i)\\}^{n}_{i=1} \\sim p(x)\\pi_0(a|x)p(s|x, a)p(r|x, a, s)$, where $s$ is short-term and $r$ is long-term reward. Under their formulation, **the two types of rewards have no difference in observation density, i.e., the long-term reward is NOT partial**. We will clarify this important distinction further in the revision.
>
> > Is there a reason why you did not include the large spectrum of pessimistic OPL?
>
> Thank you for the interesting comment. We did not discuss the pessimistic OPE/OPL because they are irrelevant in our context. It is crucial to note that our main motivation is to address the prevalent problem of partial target rewards. The techniques around pessimism do not mean to address the problem that we are tackling.  However, we could easily combine our proposed method with the pessimistic approach if one wants to do so. We will clarify this in the revision.
>
>
> > There is a problem in Equation (10), which is the core result of the paper. The gradient boils down to a doubly robust gradient that does not use any secondary reward. This is a major typo that needs fixing.
>
> Thank you for catching this. This is indeed a typo, and we have added the observational weight ($o_i/p(o_i|x_i)$) on the second line of Eq.(10) in the revision.

---

> > ### Comment · Reviewer_G83n · 2024-11-21
> >
> > I thank the authors for their answer, I would like to clarify some points.
> >
> > - **Delayed rewards should not be treated as missing rewards**. In your formulation, a delayed reward is treated the same way as a missing reward, as once we do not observe it after making the action, we get $o = 0$ and $r = N/A$. Delayed rewards are rewards that we observe, just a long time after playing an action, and requires a sequence aware framework and/or proper attribution, which is not taken into consideration by your formulation. Imagine the case of a video streaming platform, where each time the user watches a video, the system shows him a different ad for the same item. This user watches 4 videos (thus 4 ads) and later buys the item on another website. The sale is a delayed reward, it did not happen instantly and attributing it to the latest interaction/ad is suboptimal [1]. In your formulation, a delayed reward will not be taken into consideration as it is treated as missing.
> >
> > - Thank you for taking the time to conduct more experiments where $p(o|x)$ is unknown. It makes the experiments more convincing.
> >
> > > The reviewer’s example ‘using target reward when available’ would not work well unless there is a suitable function $F$, which is unknown.
> >
> > - A pretty straightforward definition of $F(s, r)$ is $F(s, r) = r$ when $r$ is available, else $F(s, r) = F(s)$. Your method combines both signals in a clever way but it needs to be compared to methods, that at least have access to both signals ($s$ and $r$) when available.
> >
> > - All the other points were addressed.
> >
> > I will raise my score after these last two points are discussed.
> >
> > [1] Fixed point label attribution for real-time bidding. Bompaire et. al.

---

> ### Author Response · Authors · 2024-11-22
>
> We would like to thank the reviewer for the additional thoughtful comments and suggestions. We will address them below.
>
> > Delayed rewards should not be treated as missing rewards. In your formulation, a delayed reward is treated the same way as a missing reward, as once we do not observe it after making the action, we get $o=0$ and $r=N/A$. Delayed rewards are rewards that we observe, just a long time after playing an action, and requires a sequence aware framework and/or proper attribution, which is not taken into consideration by your formulation. Imagine the case of a video streaming platform, where each time the user watches a video, the system shows him a different ad for the same item. This user watches 4 videos (thus 4 ads) and later buys the item on another website. The sale is a delayed reward, it did not happen instantly and attributing it to the latest interaction/ad is suboptimal [1]. In your formulation, a delayed reward will not be taken into consideration as it is treated as missing.
>
> We think this point is worth clarifying. The reviewer brought up a specific example that involves both the attribution and the delayed issues, whereas we only mentioned the latter. (Indeed, we cannot find the term “attribution” in our text and previous responses.)
>
> In our formulation and motivation, there is a logging policy $\pi_0$ that might run for a week on the platform, during which we collect data of the form $\\{(x_i, a_i, o_i, r_i)\\}\_{i=1}^n$ (excluding $s_i$ here to focus on the reviewer’s point). $x$ might include both user and video features, and $a$ represents the ad for a product that was presented. If the product has already been sold during the week, we know that $o_i = r_i = 1$. If the purchase has not yet been observed, then $o_i = 0$ and $r_i = \text{N/A}$.
>
> We think the reviewer may have conflated the attribution and delayed reward issues in this context. If the reviewer is arguing that we cannot be sure which action $a$ causes $r = 1$, that is not a problem of delay but rather a problem of “attribution” (as indicated in the title of the reference provided by the reviewer). We acknowledge that many real-world scenarios might involve both attribution and delay issues; however, **we did not claim that our formulation incorporates the attribution problem (refer to Table 1 for clarification)**. It is also worth noting that one can think of a scenario where only the issue of attribution arises without any delay, demonstrating that these two problems are orthogonal.
>
> We will explicitly discuss the distinction between attribution and delay problems in the paper to avoid any potential confusion. We would appreciate it if the reviewer could confirm whether our clarification above addresses their argument about our formulation.
>
> > A pretty straightforward definition of $F(s, r)$ is $F(s, r)=r$ when r is available, else $F(s, r)=F(s)$. Your method combines both signals in a clever way but it needs to be compared to methods, that at least have access to both signals ($s$ and $r$) when available.
>
> Thank you for the interesting suggestion. We compared the suggested baseline of using  $F(s, r)=r$ when $o=1$, and otherwise $F(s)$ with HyPeR (our proposed method). The result is summarized in the following table, which reasonably shows that our proposed method outperformed the suggested baseline (DR with $F(s, r)$) on all experimented true weights $\beta$. This observation empirically demonstrates how combining the target and secondary rewards is indeed crucial, and our proposed method does it effectively. We will add this additional result in the revised version.
>
> **Synthetic Data**
> | True Weight $\beta$            | 0.0 | 0.2 | 0.4 | 0.6 | 0.8 | 1.0 |
> |-------------------|----------|----------|----------|----------|----------|----------|
> | DR with $F(s, r)$         | 0.3203   | 0.4324   | 0.5064   | 0.5525   | 0.5419   | 0.5199   |
> | HyPeR ($\gamma=\beta$)     | 0.3940   | 0.4874   | 0.5932   | 0.6850   | 0.7338   | 0.7497   |
> | HyPeR(Tuned $\hat{\gamma}^*$) | 0.4351   | 0.5383   | 0.6262   | 0.6899   | 0.7340   | 0.7497   |
> | HyPeR(Optimal $\gamma^*$) | 0.4792   | 0.5707   | 0.6639   | 0.7261   | 0.7527   | 0.7497   |
>
> **Real-World Data**
> | True Weight $\beta$            | 0.0 | 0.2 | 0.4 | 0.6 | 0.8 | 1.0 |
> |-------------------|----------|----------|----------|----------|----------|----------|
> | DR with $F(s, r)$      | 0.2379   | 0.1937   | 0.1449   | 0.0918   | 0.0375   | -0.0143  |
> | HyPeR ($\gamma=\beta$)    | 0.2949   | 0.3027   | 0.3758   | 0.5413   | 0.6726   | 0.7833   |
> | HyPeR(Tuned $\hat{\gamma}^*$) | 0.4304   | 0.4607   | 0.5096   | 0.5753   | 0.6729   | 0.7833   |
> | HyPeR(Optimal $\gamma^*$)  | 0.5058   | 0.5381   | 0.5842   | 0.6313   | 0.6879   | 0.7833   |
>
> If the reviewer has any remaining concerns, we would be more than happy to discuss them.

---

> > ### Author Response · Authors · 2024-11-23
> >
> > We would like to inform the reviewer that we have updated our draft to clarify the points raised by the reviewers and to include additional empirical results. The updates in the text are highlighted in red. More specifically:
> >
> > - We mentioned the need for estimating the observation probabilities $p(o|x)$ in Section 4 and added experimental results with estimated probabilities $p(o|x)$ in the appendix.
> > - We added a baseline, suggested by the reviewer, of using $F(o,s, r) = o \cdot r + (1 - o) \cdot F(s)$, in the appendix.
> > - We clarified in Section 6 that, in the real-world experiment, the target and secondary rewards are not overly correlated, making it a non-trivial experiment.
> > - We empirically demonstrated the advantage of performing sampling with replacement (compared to sampling without replacement) when tuning the key parameter $\gamma$ in our method in the appendix.
> > - We mentioned the caveat of parameter tuning for OPL, as recently studied in [1].
> >
> > As the end of the discussion period is approaching, it is crucial to address any remaining concerns the reviewer may have. We would greatly appreciate it if the reviewer could confirm whether our updates have resolved their concerns or if there are any remaining points to discuss.
> >
> > [1] Saito and Nomura, 2024. Hyperparameter Optimization Can Even be Harmful in Off-Policy Learning and How to Deal with It.

---

> > > ### Comment · Reviewer_G83n · 2024-11-23
> > >
> > > Thank you for taking the time to address the majority of my concerns. I think that the paper got improved after the rebuttal and I will raise my score accordingly.
> > >
> > > On a side note, the problem of attribution is fundamentally linked to delayed rewards, especially some types of (delayed) rewards that are mentioned in Table 1. I already used the Ad-placement example and one can see a similar problem with user retention (from Table 1). This signal is mostly linked to, imagine in the case of video streaming platforms, to the whole decision system instead of an individual action (one cannot attribute the retention of a user to a recommended content). For instance, **I do not know how your current formulation can handle user retention as a delayed reward**. It can be of interest to the readers to discuss the attribution problem as a potential limitation in real life scenarios.

---

> > > > ### Author Response · Authors · 2024-11-23
> > > >
> > > > We appreciate the reviewer’s effort in carefully evaluating our work. We are sure that the reviewer’s suggestions have greatly improved our contributions, and it has been a valuable learning experience to engage in discussions with the reviewer.
> > > > We find the reviewer’s question about retention interesting, and the answer likely depends on which parts of the platform we are optimizing.
> > > >
> > > > We think the reviewer is referring to a typical recommendation scenario aimed at optimizing what content (e.g., videos) to present to users. In such a case, the formulation aligns closely with the advertisement example, because each user can be exposed to multiple or many recommendations during data collection. This scenario likely involves a mix of attribution and delay problems, which are more challenging but an interesting problem for future research.
> > > >
> > > > In contrast, we can think of bandit policies that optimize the user interface, such as the order of recommendation shelves, to use for each user for a certain period. In this case, the (potentially delayed or missing) reward is attributable to the interface chosen for the user and therefore fits into our formulation. Another example is policy optimization for email notifications, where the goal may be to decide the number of emails sent to each user per week, which can be personalized to optimize retention. This problem also aligns with our formulation, as the reward can be directly attributed to the number of emails sent.
> > > >
> > > > We totally agree that these clarifications would be useful for readers, and we will revise the draft to improve the presentation further. We would like to thank the reviewer once again for their thoughtful engagement during the discussion period.

---

### Official Review · Reviewer_nmRS · 2024-11-04

**Soundness:** 3
**Presentation:** 3
**Contribution:** 2
**Rating:** 6
**Confidence:** 3

**Summary:**

This paper studies off-policy learning with partial rewards. In the considered setting, the rewards are partially observed and the authors assume a secondary reward information.  The authors provide a weighted objective for the original and secondary reward functions based on the off-policy policy gradient algorithms. The authors then give an unbiased estimator of the gradient of the policy value and prove that the variance is reduced if the policy learned by the secondary information is better than the policy only based on the partially observed reward.  A strategy to tune the balancing weight is also given. Both synthetic and real-world datasets are provided to validate the performance.

**Strengths:**

The paper has the following strengths:
+ The paper provides an unbiased estimator for the value based on secondary information and prove that the expected gradient is the same as the one without considering secondary information ($\beta=0$).
+ The authors formally prove the variance reduction due to the introduction of the secondary information and show the reduction exists if the policy learned by the secondary information is better than the policy only based on the partially observed reward.

**Weaknesses:**

The paper can be improved in the following aspects.
- It is common to consider a weighted combination of the original and secondary information in machine learning algorithms.  Similar strategies are used in few-shot learning [2], informed learning [1], etc. The authors need to compare the proposed algorithm with them and emphasize their uniqueness.
- The paper considers the setting with some types of reinforcement learning settings and secondary information, and the authors only consider policy gradient, so I would not regard the framework as "general".
-  The strategy to tune the weight $\gamma$ is straightforward. Can the authors show the improvement in theory?







[1] Von Rueden, Laura, et al. "Informed machine learning–a taxonomy and survey of integrating prior knowledge into learning systems." IEEE Transactions on Knowledge and Data Engineering 35.1 (2021): 614-633.

[2] Wang, Yaqing, et al. "Generalizing from a few examples: A survey on few-shot learning." ACM computing surveys (csur) 53.3 (2020): 1-34.

**Questions:**

See the comments in weakness.

---

> ### Author Response · Authors · 2024-11-21
>
> We appreciate the valuable and thoughtful feedback from the reviewer. We respond to the concrete questions and comments in detail below.
>
> > It is common to consider a weighted combination of the original and secondary information in machine learning algorithms. Similar strategies are used in few-shot learning [2], informed learning [1], etc. The authors need to compare the proposed algorithm with them and emphasize their uniqueness.
>
> We thank the reviewer for providing valuable references. It is important to note, however, that **the papers and areas mentioned by the reviewer are completely orthogonal to our work regarding off-policy learning and, therefore, NOT directly comparable**.
>
> > The paper considers the setting with some types of reinforcement learning settings and secondary information, and the authors only consider policy gradient, so I would not regard the framework as "general".
>
> Thank you for pointing this out. We would like to clarify that we say our formulation is general because it can deal with _various partial observation settings_ such as delayed rewards, missing rewards, multi-stage rewards, etc, under a unified framework as summarized in Table 1. We will clarify what we mean by "general" in the revision.
>
> > The strategy to tune the weight $\gamma$ is straightforward. Can the authors show the improvement in theory?
>
> Thank you for raising this important point for discussion. While our tuning strategy might appear straightforward, this does not necessarily imply that it lacks novelty. In fact, our approach of intentionally using an incorrect (but tuned) weight $\gamma (\neq \beta)$ as opposed to the ground-truth weight $\beta$ that defines the combined policy value is, to the best of our knowledge, unprecedented. If the reviewer is aware of any similar ideas in the existing literature, we would be eager to learn about them.
>
> It is also not entirely clear what the reviewer considers to be a meaningful and interesting theoretical contribution in this context. It is important to note that the baseline here is to merely and naively setting $\gamma = \beta$ without tuning, and it is therefore expected that parameter tuning would outperform this approach. (This is analogous to comparing a predictor without any hyperparameter tuning to one with hyperparameter tuning. In this work, we propose the latter with respect to the weight in the objective function, and it is clearly more desirable.)
>
> Indeed, our experiments effectively demonstrate the advantage of weight tuning (HyPeR w/ Tuned $\hat{\gamma}$) over the naive use of the true weight (HyPeR w/ $\gamma = \beta$) across various experimental setups. Furthermore, it is worth noting that the parameter tuned via our procedure, $\hat{\gamma}$, is already nearly optimal when compared to the true optimal parameter $\gamma^*$ in terms of the combined policy value (as shown in Figures 4 and 5).

---

> > ### Comment · Reviewer_nmRS · 2024-11-23
> >
> > I thank the reviewer for the response.

---

> > > ### Author Response · Authors · 2024-11-24
> > >
> > > We greatly appreciate the timely and thoughtful discussion provided by the reviewer once again.
> > > For the reviewer’s information, we have updated our draft and included all the additional experiments suggested by the reviewer and other reviewers. More specifically:
> > >
> > > - We mentioned the need for estimating the observation probabilities $p(o|x)$ in Section 4 and added experimental results with estimated probabilities $p(o|x)$ in the appendix.
> > > - We added a baseline of using $F(s, r) = o \cdot r + (1 - o) \cdot F(s)$ in the appendix.
> > > - We clarified in Section 6 that, in the real-world experiment, the target and secondary rewards are not overly correlated, making it a non-trivial problem and crucial to effectively combine them, based on empirical evidence in the appendix.
> > > - We empirically demonstrated the advantage of performing sampling with replacement (compared to sampling without replacement) when tuning the key parameter $\gamma$ in our method in the appendix.
> > > - We mentioned the caveat of parameter tuning for OPL, as recently studied in (Saito and Nomura 2024).
> > > - We more explicitly state that our focus is on the contextual bandit setup, and that we leave a more general RL setup for future research.
> > >
> > >
> > > In our understanding, we have now addressed all the weaknesses and questions raised by the reviewers. We do not intend to challenge the reviewer’s evaluation, however, we would like to ask what prevents the reviewer from giving a clear acceptance for now, if possible. **Our main intention is to seek feedback so that we can further improve not only this submission but also our future research practices. We would also be more than happy to discuss any remaining questions regarding our work.**
> > >
> > > ---
> > > (Saito and Nomura, 2024) Hyperparameter Optimization Can Even be Harmful in Off-Policy Learning and How to Deal with It. IJCAI2024.

---

### Public Comment · ~Gholamali_Aminian1 · 2025-02-12
**Related Works**

Dear Authors,

Congrats for your great work!

In our recent work, [Semi-supervised Batch Learning From Logged Data](https://arxiv.org/pdf/2209.07148), we investigated the forward and reverse KL-regularization problem in off-policy learning with partially observed rewards. We demonstrated that the regularization term can be minimized using a partially observed dataset and achieve better performance.

We would appreciate it if you consider citing our work in your camera-ready version.

Best regards,

Gholamali Aminian

---

> ### Public Comment · ~Rikiya_Takehi1 · 2025-02-22
>
> Dear Gholamali,
>
> Thank you for your kind words and for bringing your work to our attention.
>
> We find your study relevant to our work, and we have included the citation in our paper.
>
> Best regards,
>
> Rikiya Takehi

---

### Meta-Review · Area_Chair_KZUV · 2024-12-18

**Metareview:**

This paper studies off-policy learning with partial rewards. The authors propose two approaches based on auxiliary rewards: conditioning in propensity scoring and a linear combination. The gradients of the new objectives are analyzed and the proposed methods are empirically evaluated. The scores of this paper are 4x 6, which is a major improvement over the initial 6, 2x 5, and 3. The reviewers had many concerns: typos in main results, missing related works, missing baselines, and unclear details of experiments. All of these were addressed in the discussion with the authors. The discussion was detailed and gives me confidence to suggest acceptance.

I also wanted to bring up two related works:

* [Unbiased Learning-to-Rank with Biased Feedback](https://dl.acm.org/doi/10.1145/3018661.3018699)

* [Offline Evaluation of Ranking Policies with Click Models](https://dl.acm.org/doi/10.1145/3219819.3220028)

In both, the examination probability of a position is used to counterfactually adjust the click probability of an item put at that position. This adjustment is similar to the main trick in Section 2.3.

**Additional Comments On Reviewer Discussion:**

See the meta-review for details.

---

### Decision · Program_Chairs · 2025-01-22

Accept (Poster)